# Signal transduction interfaces for field-effect transistor-based biosensors
Toshiya Sakata [1] ✉

Biosensors based on field-effect transistors (FETs) are suitable for use in miniaturized and cost-effective healthcare devices. Various semiconductive materials can be applied as FET channels for biosensing, including one- and two-dimensional materials. The signal transduction interface between the biosample and the channel of FETs plays a key role in translating electrochemical reactions into output signals, thereby capturing target ions or biomolecules. In this Review, distinctive signal transduction interfaces for FET biosensors are introduced, categorized as chemically synthesized, physically structured, and biologically induced interfaces. The Review highlights that these signal transduction interfaces are key in controlling biosensing parameters, such as specificity, selectivity, binding constant, limit of detection, signal-to-noise ratio, and biocompatibility.

A platform based on a solution-gated field-effect transistor (FET), which originates from electronics, is suitable for use in miniaturized and cost-effective systems to directly measure biological samples as the FET biosensor in the field of in vitro diagnostics[1]. Such miniaturized electronic devices can be easily equipped with a wireless function and attached to the body, which is available for wearable biosensors to detect biomarkers in tears, sweat, and saliva, that is, for diagnostics in a blood-sampling free manner[2–4]. In general, the gate insulator surface (e.g., $SiO_2$) is directly in contact with a measurement solution in the FET biosensor without a metal gate electrode, which is different from a metal-oxide-semiconductor (MOS) transistor, for which the potential of the measurement solution is controlled by the reference electrode[5–7], as shown in Fig. 1a. When ions or biomolecules with charges are adsorbed on the gate insulator surface, their charges electrostatically interact with electrons across the gate insulator, resulting in a change in the conductivity of the channel of the FET. That is, the drain–source current ($I_{DS}$) at the channel changes with the change in the density of ions or biomolecules with charges adsorbed on the gate insulator. That is, such charged species induce a change in the potential ($V_{out}$) of the gate insulator/electrolyte solution interface at a constant $I_{DS}$, which is potentiometrically detected[5,8]. Oxide and nitride membranes (e.g., $Ta_2O_5$ and $Si_3N_4$) used as the gate insulator contribute to the detection of a change in pH with the FET on the basis of the reaction in equilibrium between hydrogen ions and hydroxy groups at their surfaces[5]. Afterwards, various ion-sensitive membranes (ISMs)[2,9–15] and biological receptors with enzymes, antibodies, and single-stranded DNAs[16–21] were modified on the gate electrode of the FET biosensors to specifically and selectively detect target ions and biomolecules, and further cellular activities, which were accompanied by changes in ion concentration, were monitored in real-time with the FET biosensors[22–25].

Thus, the detection principle of FET biosensors is based on the potentiometric measurement of the changes in ionic and biomolecular charges or membrane capacitances at the gate electrode/electrolyte solution interface. Moreover, one-dimensional (1D) and two-dimensional (2D) semiconductive materials have been recently proposed for the channel of FETs for biosensing devices[15,26–30]. In particular, a solution-gated 1D or 2D-channel FET biosensor with a steep subthreshold slope (SS) contributes to an ultrahigh sensitive biosensing, owing to a relatively large electric double-layer capacitance at the electrolyte solution/channel interface[27,31]. Moreover, thin-film transistors with transparent amorphous oxide semiconductors can be applied as one of the FET biosensors, which are deposited on transparent substrates such as glass and plastics[32]. Thus, the number of studies on the FET biosensors is increasing yearly (Fig. 1b).

Considering the concept of biosensors, which consist of three components, namely, a biological sample, a signal transduction interface, and a detection device (Fig. 1a)[1,33], a specific biological target detected should be considered in developing FET biosensors for their applications in various fields such as clinical diagnostics and pharmaceutical discovery. In particular, the signal transduction interface is designed and positioned between the biological sample and the detection device to specifically and selectively detect the target biomarker. Functionalized interfaces are chemically synthesized, physically and chemically structured, and biologically induced, considering the diversity of material properties (Fig. 1a). Moreover, as a detection device, various sensing principles such as electrochemical and optical methods are proposed for the detection of biomarkers; particularly, the solution-gated FETs for biosensing are focused on in this review, considering the topical backgrounds mentioned above.

[1]Department of Materials Engineering, School of Engineering, The University of Tokyo, 7-3-1 Hongo, Bunkyo-ku, Tokyo 113-8656, Japan.
✉e-mail: sakata@biofet.t.u-tokyo.ac.jp

**Fig. 1 | Application of FET for biosensing.**
**a** Conceptual structure of potentiometric FET biosensor, which consists of three components, namely, a biological sample, a signal transduction interface, and a detection device. **b** Yearly number and cumulative number of publications on FET biosensors obtained by a search in PubMed, where "FET biosensor", "bio-FET", or "biotransistor" was used as a keyword.

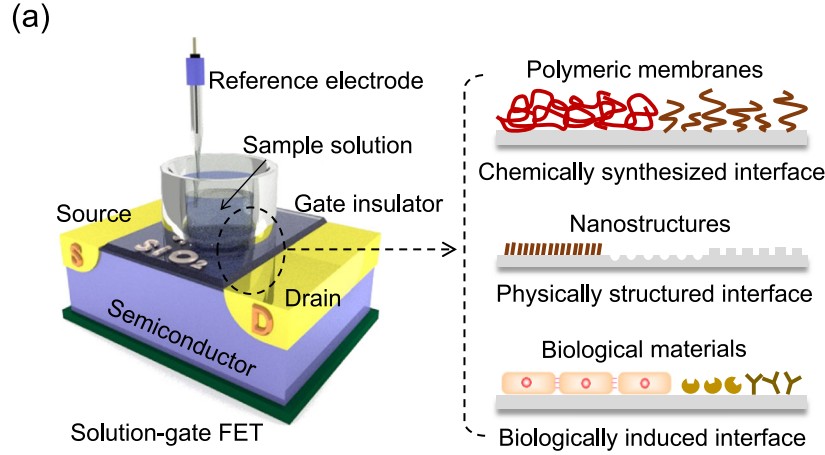

(a)

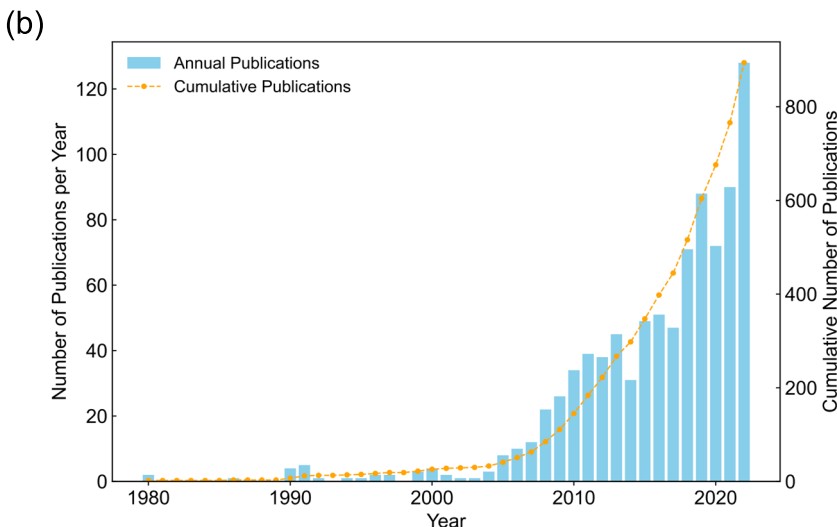

(b)

In electrochemical biosensors such as FET biosensors, the signal transduction interface between the electrode surface and the biological sample plays a key role in capturing target ions or biomolecules, which then transduce the electrical properties of ions or biomolecules and electrochemical reactions into output signals (Fig. 1a). In general, biological receptors such as enzymes and antibodies are often utilized on the electrode as a biologically induced electrical interface to transduce their reactions with target biomarkers into electrical signals, showing high specificity and selectivity in their detections[16,34–37]. However, the use of such biological materials is problematic because there are no enzymes or antibodies applicable to every target biomarker, as well as other reasons such as their poor long-term stability, high cost and time-consuming production, and the difficulty of quality control of their production. On the other hand, polymer membranes such as molecularly imprinted polymers (MIPs)[38–43] and aptamers[44–46] that are artificially designed and simply fabricated are expected as a platform of the signal transduction interfaces of FET biosensors, thereby not only overcoming the above issues of biological materials, but also enhancing the binding constant ($K_a$) and thereby the limit of detection (LOD). These polymer membranes are arranged on the gate electrode surface as a chemically synthesized electrical interface[43,47–55]. Such flexible polymeric membranes are also available for wearable biosensors. That is, it is better to use soft materials such as polymeric membranes or a thin film as the signal transduction interfaces to decrease the stiffness of the wearable and flexible biosensors. Moreover, living cells themselves, which are cultured on the gate electrode surface of FET biosensors, can also work as a biologically induced electrical interface. For instance, mast cells with IgE antibodies cultured on the gate electrode induce electrical signals on the basis of the change in pH, which is caused by the immunological reactions of the antibodies with the target antigen[56,57]. Moreover, vascular endothelial cells with a basement membrane cultured on the gate electrode are induced to invade cancer cells, resulting in the generation of electrical signals of FET biosensors[58,59]. Such living cells with distinctive functions can be utilized as one of the biologically induced electrical interfaces similar to immobilized enzyme membranes. Thus, we can design diverse signal transduction interfaces for electrochemical biosensing at not only the molecular level but also the cellular level.

However, we have to address concerns on how to increase the signal-to-noise ratio (S/N) for the FET biosensors, based on not only the effect of counterions in sample solutions capable of shielding targeted biomolecular charges (Debye length limitation) thereby reducing the output signals[18,20,60–68], but also noise signals that are unexpectedly caused by the fouling and nonspecific adsorptions of interfering species in biological sample solutions[69–73]. The Debye length $\lambda$ depends on the ionic strength of the electrolyte solution used and is expressed as $\lambda = (\varepsilon_0 \varepsilon_r k_B T / 2 N_A e^2 I)^{1/2}$, where $I$ is the ionic strength of the electrolyte solution, $\varepsilon_0$ is the permittivity of free space, $\varepsilon_r$ is the dielectric constant, $k_B$ is the Boltzmann constant, $T$ is the absolute temperature, $N_A$ is the Avogadro number, and $e$ is the elementary charge. The Debye length limitation is controlled by changing the ionic strength of a measurement solution, that is, diluted measurement solutions are useful for improving the detection sensitivity of the FET biosensors to charged biomolecules because of the reduction of the shielding effect by counterions. Furthermore, in terms of increasing S/N, convex (e.g., nanopillars)[74–76] and concave (e.g., nanofilters and nanopores)[70–73,75] electrode surfaces may be effective as physically and chemically structured

**Fig. 2 | Polymeric ISM as a chemically synthesized electrical interface. a** Most popular ISM. The change in the density of ionic charges captured on/in the ISMs under equilibrium contributes to the change in the interfacial potential at the gate electrode surface. Organic solvents such as plasticizers and ionophores that leak into biological samples may cause cytotoxicity. **b** Noncytotoxic plasticizer-free ISM. **c** Anti-biofouling on ISM realized by coating hydrophilic polymer.

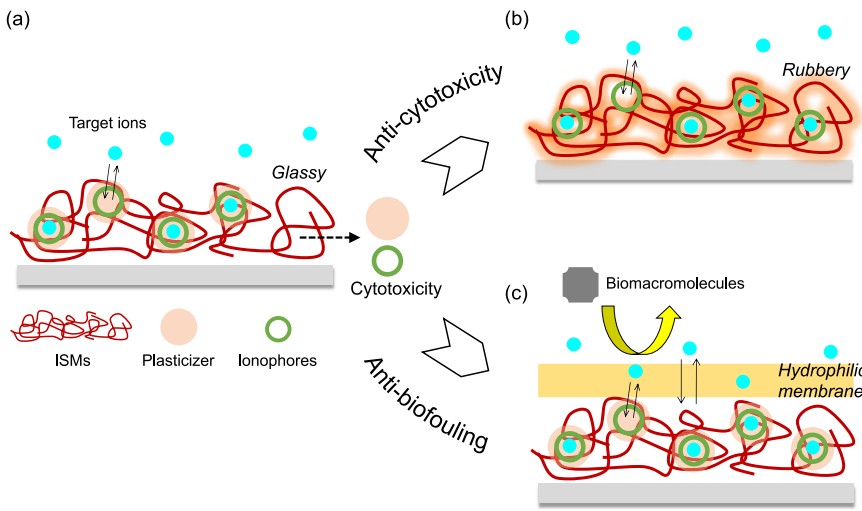

electrical interfaces. The increase in electrode surface area is expected to enhance the output signals of FET biosensors, whereas it may generate noise signals simultaneously. In this case, the electrical noise signals can be suppressed in the FET biosensors when small and large interfering species are trapped outside the Debye length by a polymeric nanofilter interface, blocking their approach to the electrode surface, which means the suppression of nonspecific noise signals. That is, the polymeric nanofilter structurally leads electrical signals to a small target biomarker, which passes through the nanofilter interface to the electrode while preventing small interfering species from approaching the electrode[70–72]; that is, S/N can be increased.

Considering the application of electric devices with FETs to biosensors, it is very important to design devices such that the change in the charge density based on biomolecular recognition events is directly transduced into electrical signals at the signal transduction interfaces without relying on enzymes and antibodies. Therefore, the signal transduction interfaces classified as chemically synthesized, physically and chemically structured, and biologically induced interfaces become a key element in controlling the performances of FET biosensors, which determine their future applications. However, there are no reports on the signal transduction interfaces as classified in the above. In this paper, we review some distinctive signal transduction interfaces classified in relation to FET biosensors, which transduce biomolecular recognition events into electrical signals. In particular, the diverse signal transduction interfaces in FET biosensors are noted to become a key element in controlling biosensing parameters with respect to biological targets, such as specificity, selectivity, $K_a$, LOD, S/N, and biocompatibility. Therefore, the diversity of signal transduction interfaces should broaden the possibility of developing novel biosensing devices, in parallel with the development of new channel semiconductors of FET biosensors.

## Chemically synthesized electrical interfaces for FET-based biosensing

Polymeric membranes can be artificially functionalized by copolymerizing characteristic monomers and entrapping distinctive chemicals in a biological receptor-free manner. In particular, functional polymeric membranes should have the ability to transduce the interactions of such chemicals with charged ions and biomolecules into electrical signals. The chemically synthesized electrical interfaces must be accordingly designed, considering the physicochemical characteristics and components of polymeric membranes. As the typical ISMs, hydrophobic polymers are mostly utilized to entrap poorly water-soluble ionophores such as crown ethers that coordinate ions with charges, which dissolve in a plasticizer[2,9–15], whereas water-soluble biomolecules such as glucose are easily incorporated into hydrophilic

polymeric membranes with functional monomers such as phenylboronic acids (PBAs), which expectedly induce electrical charges on the basis of diol binding to some biomolecules owing to the formation of deprotonated boronic acid diol esters with negative charges[49–52,77,78]. These chemical interactions, which contribute to the change in the density of ionic and molecular charges, on/in such polymeric membranes, are output as the electrical signals of FET biosensors.

### Ion-sensitive membranes and their biocompatibility

Most popular ion sensors are composed of hydrophobic and flexible polymeric membranes such as polyvinyl chloride (PVC), including ionophores coated on electrodes for the potentiometric measurements of target ions. Artificial ionophores such as crown ethers and calixarenes are dissolved in a plasticizer to enhance their mobilities in the flexible PVC membrane. Various ions such as $Na^+$, $K^+$, $Ca^{2+}$, $NH_4^+$, and $Cl^-$ can be detected using the potentiometric ISM-coated electrodes, including FETs (Fig. 2a)[2,9–15]. Thus, flexible ISMs enable the detection of inorganic ions in biological and environmental samples. The difference in the potential at the ISM/electrolyte solution interface ($\Delta E$) changes with the change in ion concentrations, that is, the density of ionic charges captured on/in the ISMs under equilibrium. That is, the detection sensitivity of potentiometric sensors with ISMs ideally follows the slope obtained on the basis of the Nernstian equation (e.g., 59.2/$n$ mV/decade for $n$-valent ions at 298 K). However, the detection sensitivity may be lower, depending on the density of ionophores entrapped in the ISMs. Here, $\Delta E$ is expressed as Eqs. 1 and 2 on the basis of the Nernstian equation in relation to the logarithm of the $n$-valent cation concentration [$M^{n+}$]:[7]

$$\Delta E = 2.303 \frac{kT}{nq} \left( \frac{\beta}{\beta + 1} \right) \Delta \log [M^{n+}], \qquad (1)$$

$$\beta = \frac{2q^2 N_S K_a^{1/2}}{kT C_{DL}}, \qquad (2)$$

where $k$ is the Boltzmann constant, $T$ is the absolute temperature, $q$ is the elementary charge, and $C_{DL}$ is the capacitance of the electric double layer. The parameter $\beta$ reflects the chemical sensitivity of ISMs, which depends on the site density $N_S$ of ionophores, and the surface reactivity is characterized on the basis of $K_a$ of the ionophore–ion coordination bond. Thus, the site density of ionophores is a major factor that determines $\Delta E$, that is, the detection sensitivity of ISMs. Moreover, the selectivity of a crown ether L for ions $M_1^{n+}$ and $M_2^{n+}$ is expressed as the ratio of each binding constant $K_a(1)$/$K_a(2)$; e.g., $K_a(1) = [M_1 L^{n+}]/([M_1^{n+}] + [L])$. The complexing ability, that is, $K_a$ of crown ether with ions, is dependent on the relative sizes of the cavity of

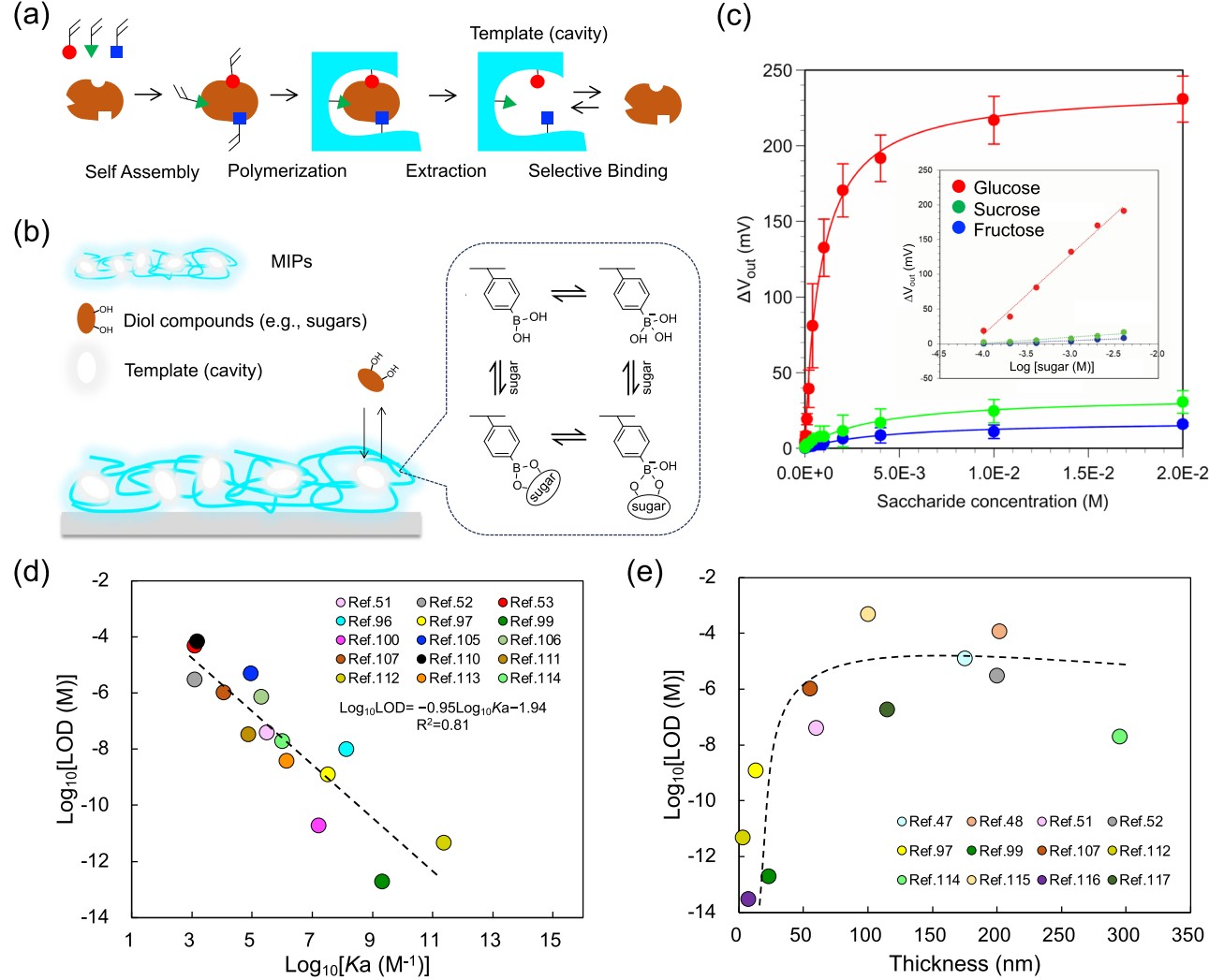

**Fig. 3 | MIP-based chemically synthesized electrical interface. a** Fabrication process of MIP. **b** An example of MIP coated on the gate electrode surface. PBA is attractive as one of the functional molecules for inducing the binding to diol compounds, resulting in the change in the density of molecular charges. **c** Change in the interfacial potential of the glucose-selective MIP-based FET for each sugar concentration (red, glucose; blue, fructose; and green, sucrose). Plots were approximated by the Langmuir absorption isotherm. Error bars represent the standard deviation determined from the number of experiments ($n = 3$ at each concentration). Credit: from ref. 52. Reprinted with permission from the American Chemical Society. **d** Correlation between $K$a and LOD of MIPs based on the data shown in fifteen previous papers. **e** Correlation between thickness and LOD of MIPs based on the data shown in 12 previous papers.

the crown ether and the ions[79]. In addition, poly- and bis (crown ether)s are favorable for the formation of 2:1 (crown ether unit/ion) sandwich-type complexes with ions, which contribute to the increase in $K_a$, that is, the selectivity to a specific ion[80].

Moreover, the biocompatibilities of ISMs with flexibility should be considered for wearable biosensors, focusing on cytotoxicity and biofouling. Plasticizers used in ISMs may be cytotoxic when used in direct contact with the human body as a wearable biosensor. In fact, plasticizers were found to be cytotoxic to living cells[13]. The plasticizers and ionophores included in ISMs may leak into the sample solution and then result in it becoming cytotoxic (Fig. 2a). That is, we had better develop a plasticizer-free ISM, from which the ionophores do not leak, to apply to biocompatible ion sensors. In this case, the mobility and dispersibility of ionophores in a plasticizer-free polymeric material should be considered, and then the matrix material that has polarity and a glass transition temperature below room temperature is preferred. Moreover, the dielectric constant of the polymer selected is desired to be in the range of 4–15 so as not to increase membrane resistance [e.g., fluoropolysilicone (FPS)][81,82]. A plasticizer-free $Na^+$-sensitive FPS-gate FET with calix[4]arene actually exhibited higher biocompatibility (i.e., noncytotoxicity) and sufficient sensitivity to and selectivity for $Na^+$, as

expected (Fig. 2b)[83]. On the other hand, we have another issue with ISMs that show nonspecific adsorption of, for example, proteins in whole samples such as blood, that is, biofouling of ISMs, resulting in the reduction in sensitivity. As an anti-biofouling treatment, ISMs are modified with some hydrophilic polymers (Fig. 2c)[84,85]. Hydrophilic polydopamine (poly-DA) modification actually enhanced the anti-adhesive properties of ISMs by increasing the surface hydrophilicity[85]. This is why a poly-DA film prevented the nonspecific adsorptions of proteins that may generate noise signals on the ISMs, while the detection sensitivity for monovalent ions did not deteriorate, keeping it near the Nernstian response. Thus, the modified ISM-based FETs can be utilized more safely and precisely in actual biological samples with interfering species as wearable biosensors with flexibility.

## MIP-based electrical interfaces

As described in Introduction, there are some issues with biological receptors such as enzymes and antibodies; therefore, there is a need to develop versatile polymeric membranes to capture various biomarkers as intended. In the case of FET biosensors, such functional polymeric membranes should induce electrical signals. An MIP is a soft and biomimetic material developed to artificially realize the selective detection of targets[38–42]. A rigid and

highly crosslinked polymer matrix contributes to the selective molecular recognition, which generates a strong target–functional monomer interaction (Fig. 3a). Thanks to their simple preparation and versatility, molecular imprinting technologies have been applied to various biosensing devices to selectively detect targets[86–89]. In particular, MIP films have been modified onto the gate electrode of FET biosensors for the specific and selective detection of small biomolecules to form a chemically synthesized electrical interface[43,47–53]. PBA is often utilized as the target-interacting functional monomer for a MIP-based electrical interface for the FET biosensor (Fig. 3b)[43,49–53], because it covalently forms stable esters with diol biomolecules (e.g., saccharides, catechol amines, and lactic acid), which are water-soluble chemicals (Fig. 3b). This is why PBA has attracted a lot of attention in the field of molecular recognition. Furthermore, the esterification of PBA–diol compound binding is based on a reversible reaction, depending on pH[78]. Therefore, the template molecules can be eluted from the polymerized matrix by simply controlling pH. In particular, PBA is in equilibrium at an anionic form in the esterification (Fig. 3b). For instance, glucose molecules used as the template, which bind to PBA, are removed under acidic conditions, whereas glucose boronate esters induce negative charges in the MIP matrices under relatively basic conditions because the pKa of a glucose boronate ester is 6.8[78]. Hence, FET biosensors can detect the change in the density of negative charges generated by the PBA–diol compound binding on the basis of the detection principle.

Here, a quantitative analysis is needed to grasp the chemical basis of MIP–target interactions underlying the electrical responses of MIP-based FET biosensors. The binding characteristics of a target to MIP are often quantified with adsorption isotherm equations in the case that the binding process involves the reversible reaction of the target to the target-selective MIP. According to a previous study[90], the Langmuir adsorption isotherm equation is described as

$$A = \frac{N[c]}{1 + K_a[c]}, \tag{3}$$

where $A$ is a signal observed at equilibrium for the MIP-bound template, $[c]$ the free concentration of the template at equilibrium, $N$ the number of available active centers in the MIP per unit volume, and $K_a$ the binding constant. Equation 1 assumes homogeneously distributed binding sites with a constant binding constant $K_a$.

In the unsaturated region, the operation of a silicon-based FET is represented as

$$I_{DS} = \mu C_{OX} \frac{W}{L} \left[ (V_{GS} - V_T)V_{DS} - \frac{1}{2} V_{DS}^2 \right], \tag{4}$$

where $\mu$ is the electron mobility in the channel, $C_{OX}$ is the gate oxide capacitance, $\frac{W}{L}$ is the channel width-to-length ratio, $V_{DS}$ and $V_{GS}$ are the applied drain–source and gate–source voltages, respectively, and $V_T$ is the threshold voltage, which is described as ref. 7

$$V_T = E_{ref} - \psi_0 + \chi^{sol} - \frac{\phi_{si}}{q} - \frac{Q_{it} + Q_f + Q_B}{C_{OX}} + 2\phi_f, \tag{5}$$

where $E_{ref}$ refers to the potential of a reference electrode relative to a vacuum, $(-\psi_0 + \chi^{sol})$ to the interfacial potential between the electrolyte solution and the gate oxide electrode (the factor $\chi^{sol}$ is the surface dipole moment of the solution, which can be considered constant), $\frac{\phi_{si}}{q}$ to the work function for silicon electron, $Q_{it}$, $Q_f$ and $Q_B$ to the charge of the interface traps, the fixed oxide charge, and the bulk depletion charge per unit area, respectively, and $\phi_f$ to the Fermi potential difference between the doped bulk silicon and the intrinsic silicon.

When the MIP film is coated on the gate oxide electrode of the FET, the capacitance and charge in the MIP film should be added to Eq. 3, which is

modified as

$$V_T = E_{ref} - \psi_0 + \chi^{sol} - \frac{\phi_{si}}{q} - \frac{Q_{it} + Q_f + Q_B + Q_{MIP}}{C_{Com}} + 2\phi_f \tag{6}$$

with

$$C_{Com} = \frac{C_{OX} \cdot C_{MIP}}{C_{OX} + C_{MIP}} = \frac{C_{OX}}{1 + \frac{C_{OX}}{C_{MIP}}}, \tag{7}$$

where $Q_{MIP}$ is the charge of the MIP film, and $C_{Com}$ is the combined capacitance of $C_{OX}$ and the MIP film ($C_{MIP}$) on the gate oxide electrode. Considering that $C_{MIP}$ would hardly change after the addition of targeted molecules[43,91] and $C_{OX}$ is also constant, $C_{Com}$ is almost constant regardless of the adsorption of biomolecules, especially small molecules. In addition, the change in $\psi_0$ is assumed to remain unchanged because pH can be basically maintained by using a buffer solution. $E_{ref}$, $\frac{\phi_{si}}{q}$, $Q_{it}$, $Q_f$, $Q_B$, and $\phi_f$ are the factors that do not change even after molecular recognition events at the MIP interface. Considering the above, the electrical response output with the MIP-based FET biosensor, that is, the change in $V_T$ ($\Delta V_T$) is involved in $\Delta Q_{MIP}$ on the basis of Eq. 4.

The binding affinity of PBA to a diol compound depends on pH in a solution, in particular, the B(OH)$_3^-$ complex is much stabler than the B(OH)$_2$ complex[92]. That is, the reversible reaction between the target diol biomolecule (T) and PBA in the MIP film is described as

$$T + PBA \leftrightarrow T \cdot PBA^-, \tag{8}$$

the formation rate of the T · PBA$^-$ complex at time $t$ is written as

$$\frac{d[T \cdot PBA^-]}{dt} = k_a[T][PBA] - k_d[T \cdot PBA^-], \tag{9}$$

where $k_a$ is the association rate constant and $k_d$ is the dissociation rate constant. At time $t$, $[PBA] = [PBA]_0 - [T \cdot PBA^-]$, where $[PBA]_0$ is the concentration of PBA at $t = 0$. This is substituted into Eq. 7 to give

$$\frac{d[T \cdot PBA^-]}{dt} = k_a[T]([PBA]_0 - [T \cdot PBA^-]) - k_d[T \cdot PBA^-]. \tag{10}$$

Here, the charge density of the MIP film ($Q_{MIP}$) is proportional to the formation of the T · PBA$^-$ complex in the MIP film. Then, the maximum $Q_{MIP}$, that is, $Q_{max}$ is dependent on the PBA concentration in the MIP film ($[PBA]_0$ at $t = 0$), which is based on the capacity of the incorporated ligand. Therefore, Eq. 8 is modified to

$$\frac{dQ_{MIP}}{dt} = k_a[c](Q_{max} - Q_{MIP}) - k_d Q_{MIP} = k_a[c]Q_{max} - (k_a[c] + k_d)Q_{MIP}, \tag{11}$$

where $\frac{dQ_{MIP}}{dt}$ is the formation rate of the associated complex (T · PBA$^-$) in the MIP film on the gate oxide surface and $[c]$ is the concentration of the analyte (T) in a solution. In addition, integrating Eq. 9 gives

$$Q_{MIP}^t = \frac{k_a[c]Q_{max}[1 - e^{-((k_a[c]+k_d)t)}]}{k_a[c] + k_d} = \frac{[c]Q_{max}}{[c] + 1/K_a}[1 - e^{-((K_a[c]+1)t)}], \tag{12}$$

where $K_a$ is the binding constant of T and PBA ($k_a/k_d$) in the MIP film. From Eq. 10, $Q_{MIP}^{t=0} = 0$. Considering Eq. 4,

$$\Delta V_T(-\Delta V_{out}) = -\frac{\Delta Q_{MIP}^t}{C_{Com}} = -\frac{[c]\Delta V_{out}^{max}}{[c] + \frac{1}{K_a}}[1 - e^{-((K_a[c]+1)t)}] \approx -\frac{[c]\Delta V_{out}^{max}}{[c] + 1/K_a}, \tag{13}$$

which is estimated at an equilibrium time $t$. Here, $\Delta V_{out}^{max}$ is the maximum change in potential induced by $\Delta Q_{max}$ at the MIP interface, which is proportional to the density of binding sites. Actually, $\Delta V_{out}$ at the MIP interface is measured as the change in the surface potential against the reference electrode at a constant $I_{DS}$ using the source follower circuit[8], as shown in Fig. 3c. That is, the detected $\Delta V_{out}$, which is induced by $\Delta Q$, is consistent with $-\Delta V_T$ obtained in the $V_{GS}$ and $I_{DS}$ transfer characteristics.

Considering the above, the electrical signals of MIP-based FET biosensors obey the Langmuir adsorption model. Therefore, the adsorption isotherm equation for the MIP-FET system is obtained from Eqs. 1 and 11 as

$$\Delta V_{out} = \frac{\Delta V_{out}^{max}[c]}{1/K_a + [c]}, \qquad (14)$$

where $[c]$ is the target concentration at equilibrium. Furthermore, the homogeneity and heterogeneity of binding sites distributed in MIPs are critical to the effective increase in selectivity. The binding sites in MIPs are assumed to be heterogeneously distributed owing to the randomness of polymerization and the insufficient elution of template molecules from the matrixes, depending on the polymerization processes[93]; that is, MIPs may unintentionally contain both nonselective and highly selective binding sites at a certain ratio. In that case, the bi-Langmuir adsorption isotherm equation (Eq. 13) may be utilized for the heterogeneous binding model of the MIP-FET system, instead of Eq. 12.

$$\Delta V_{out} = \frac{\Delta V_{1\_out}^{max}[c]}{1/K_{a1} + [c]} + \frac{\Delta V_{2\_out}^{max}[c]}{1/K_{a2} + [c]} \qquad (15)$$

Equation 13 assumes two main types of binding sites with different affinities in the heterogeneous MIP film. However, $K_a$ can be simply compared among MIPs for various target biomolecules using Eq. 12, as far as the results analyzed in most previous works are concerned.

In addition, stiffer MIP matrices may be more selective; that is, it should be beneficial to prepare flexible polymeric membranes as rigid as possible[94]. Actually, electroactive polymeric membranes with relatively high Young's moduli, such as polypyrrole (PPy) and poly(o-phenylenediamine) (PoPDA) appear to be utilized as the MIP matrices, preserving the macromolecular arrangements after the template extraction[95–101]. On the other hand, 2-hydroxyethylmethacrylate (HEMA), ethylene glycol dimethacrylate (EGDMA), and so forth are copolymerized to improve the hydrophilicity of MIP matrices with flexibility, which include small biomolecules and electrolytes and may show relatively low Young's moduli, when PBA derivatives (e.g., 4-vinyl-PBA) are copolymerized in the MIP matrices on the gate electrode (Fig. 3b)[43,49–52]. Such hydrophilic MIPs contribute to the reduction in the intensity of noise signals based on the nonspecific adsorptions of interfering biomacromolecules such as proteins in sample solutions[102]. Moreover, basic N-[3-(dimethylamino)propyl]methacrylamide (DMAPM) is often incorporated in the MIP matrices, including PBA to control pH[43,49–52,103]. Furthermore, methacrylic acid (MAA) can be used as a functional monomer in the MIP matrices because of its versatility in interactions[43,50,53,104–107]. MAA is used to electrostatically interact with amino groups in target biomolecules, to further increase the selectivity of biosensors. Note that MIP matrices should be crosslinked to maintain the cavity size and shape, which are associated with the specificity and selectivity for target biomolecules. The rigidity of MIPs would also block nontarget biomacromolecules such as proteins from coming in contact with the gate electrode, which contributes to the suppression of nonspecific signals. In particular, the swelling–deswelling behavior of such hydrogels after the reaction with target biomolecules may cause the change in the capacitance of MIP films and then cancel out the electrical signals on the basis of the change in the density of molecular charges. Therefore, the density of cross-linking in the MIP matrices on the gate electrode should be controlled to minimize the swelling–deswelling behavior[52,91], although such capacitive signals may be useful for the detection of target biomolecules. Actually, a glucose-selective MIP, which was randomly copolymerized as poly(HEMA-ran-DMAPM-

ran-VPBA-ran-MAA) and cross-linked with MBA on the gate surface of FET, worked well as a chemically synthesized electrical interface of FET biosensors (Figs. 3b, c)[52]. At pH 7.4, the binding constant for glucose ($K_a^{glucose}$) with the glucose-selective MIP-FET was estimated as $1.2 \times 10^3\,M^{-1}$ by fitting the Langmuir isotherm equation (Fig. 3c), which was approximately 260 times higher than that of the pristine PBA–glucose binding ($4.6\,M^{-1}$), whereas the binding constant for fructose ($K_a^{fructose}$) in the glucose-selective MIP ($2.2 \times 10^2\,M^{-1}$) hardly changed from that for the pristine PBA–fructose binding ($1.6 \times 10^2\,M^{-1}$)[78]. This means that not only did the imprinting for the template molecule in the polymeric membrane increase $K_a$ for the target biomolecule, but the molecular charges induced by the diol compound/PBA binding also contributed to the generation of electrical signals of the FET biosensor. In this case, the selectivity (S) for glucose in the MIP film was calculated from the ratio of $K_a^{glucose}$ to the $K_a$ of other sugars and PBA ($K_a^{sugars}$), that is, $S_{glucose/fructose}^{MIP} = K_a^{glucose}/K_a^{fructose} = 5.6$ [52]. From these calculations, the detection selectivity for glucose to fructose using the glucose-selective MIP-FET was about 200 times higher than that in the pristine PBA–sugar binding ($S_{glucose/fructose}^{pristine} = 2.9 \times 10^{-2}$). In this glucose-selective MIP-FET, moreover, LOD was determined to be ~3 μM for glucose detection in accordance with the Kaiser limit theory[108], which can be compared with that in glucose-responsive MIPs obtained by other readout technologies[109]. Here, LOD would be associated with $K_a$ in Eq. 12. When $\Delta V^{out}$ at the concentration $[c]_{LOD}$ is $\Delta V_{out}^{LOD}$, Eq. 12 is modified to

$$\Delta V_{out}^{LOD} = \frac{\Delta V_{out}^{max}[c]_{LOD}}{1/K_a + [c]_{LOD}}. \qquad (16)$$

Then, Eq. 14 is rearranged as

$$\log_{10}[c]_{LOD} = -\log_{10}K_a + \log_{10}\frac{\Delta V_{out}^{LOD}}{\Delta V_{out}^{max} - \Delta V_{out}^{LOD}}. \qquad (17)$$

Actually, the glucose-selective MIP-FET shown above sufficiently satisfies Eq. 15. Furthermore, this trend can also be found in other MIPs, regardless of the readout technologies (Fig. 3d)[51–53,96,97,99,100,105–107,110–114]. Therefore, $K_a$ is a parameter used to control LOD. Note that $K_a$ of the target/MIP interaction may be inherently derived from that of the target/functional monomer (e.g., PBA–diol compounds and host–guest interactions). Moreover, as another parameter, the thickness of MIP films appears to be related to the LOD (Fig. 3e)[47,48,51,52,97,99,107,112,114–117]. Thin-film MIPs with a thickness less than ca. 50 nm would contribute to the enhancement of LOD, which may result from a high $K_a$. This means that thicker MIP films probably include more heterogeneous sites for binding to target biomolecules[50]. The thickness and adhesiveness of MIP films at substrates can be precisely controlled to make them thinner (<50 nm) by some grafting methods, such as surface-initiated atom transfer radical polymerization (SI-ATRP), which results in a higher $K_a$[51,112].

In contrast, a nonimprinted polymer (NIP) should be synthesized on the gate electrode as a control polymer by the same method as that for MIP except for adding a template molecule. Even NIPs may show some affinities with target biomolecules owing to their nonspecific adsorptions and other properties, resulting in a high $K_a$. The imprinting factor (IF) is often evaluated as one of the imprinting parameters for target biomolecules, $IF = K_a^{MIP}/K_a^{NIP}$[47,51,94,96,101,105,109,111,114,115]. That is, a higher IF indicates better MIP performance. For instance, a non-glucose-selective NIP-FET against the glucose-selective MIP-FET mentioned above hardly showed glucose responsivity within the relatively wide range of concentrations, the $K_a^{NIP}$ of which was difficult to analyze[52]; that is, $K_a^{NIP}$ was low, resulting in a very high IF. This may be because the NIP suppresses not only the uptake of glucose molecules into itself owing to the higher cross-linking density but also the nonspecific adsorptions owing to its hydrophilicity based on HEMA. Note that a PBA-containing hydrogel-coated FET with a lower cross-linking density, the composition of which was almost the same as that of the non-glucose-selective NIP-FET, showed some electrical signals for the change in

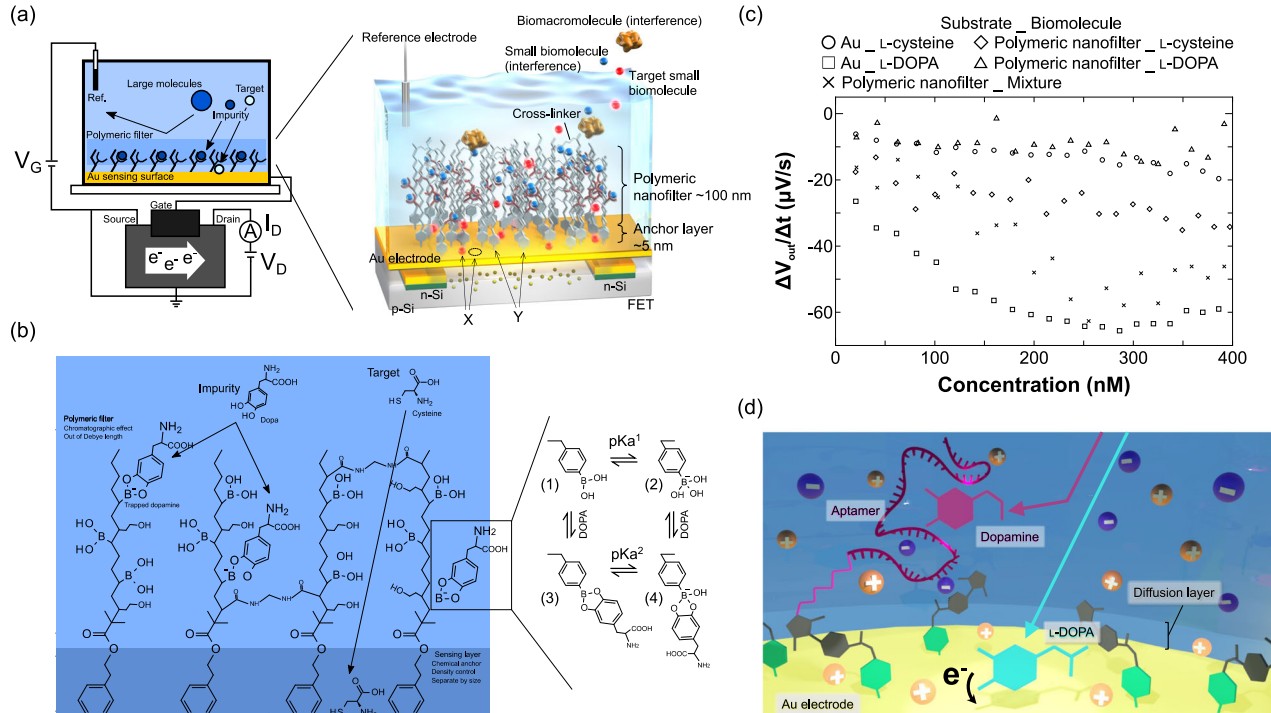

**Fig. 4 | Polymeric nanofilter as chemically and physically structured interface.**
**a** Conceptual illustration of EG-Au-FET biosensor with nanofilter interface. Conceptual design of nanofilter interface to trap interfering small biomolecules and specifically detect target small biomolecules by Au electrode. **b** Conceptual design of nanofilter interface to trap L-DOPA and specifically detect cysteine by Au electrode. **c** Reaction rates for L-cysteine and L-DOPA interacting with the Au electrode using the EG-Au-FET with or without the polymeric filter. These reaction rates were calculated on the basis of the data obtained in a previous work[70]. L-Cysteine was added onto the unmodified Au electrode (○) or the polymeric nanofilter-coated Au electrode (◊), L-DOPA was added onto the unmodified Au electrode (□) or the polymeric nanofilter-coated Au electrode (△), and their mixture was added onto the polymeric nanofilter-coated Au electrode (×). In this figure, all the polymeric nanofilters contained PBA receptors. Credit in **a–c**. From ref, 70. Reprinted with permission from the American Chemical Society. **d** Conceptual structure of aptamer-based nanofilter interface. Credit: From ref. 55. Reprinted with permission from Elsevier.

glucose concentrations[91]. Thus, the $K_a$ values of MIPs and NIPs are the important parameters for controlling $S$, LOD and IF in the chemically synthesized electrical interface. The number of studies on MIPs is increasing yearly (Figure S1); their further applications to FET biosensors are expected in the future, the number of which is still small (Figure S2).

In this section, the chemically synthesized interfaces have been introduced to enhance the performance of FET biosensors, focusing on ISMs and MIPs. Functional compounds such as crown ether and PBA should be included in such polymeric membranes to induce the specificity of targets. In addition, poly- and bis (crown ether)s and the molecular imprinting increase $K_a$, which should result in a lower LOD and a higher $S$ for targets. Note that such chemically synthesized interface materials should be designed with consideration for biocompatibility when used in direct contact with the human body as a wearable biosensor.

## Physically and chemically structured electrical interfaces for FET-based biosensing

Nanostructured electrodes with nanopillars and nanopores may be effective for increasing S/N for a physically structured electrical interface (Fig. 1a). The increase in the surface area of a nanostructured electrode is expected to increase the intensity of output signals, but may also generate noise signals simultaneously. However, a previous paper showed that electrostatic screening is weaker in the vicinity of concave surfaces and stronger in the vicinity of convex surfaces[75]. That is, molecular charges tethered at the convex gate surface (e.g., nanopillars) are shielded by counterions in a sample solution, and thus no electrical signals may be induced in the FET biosensors. On the other hand, porous membranes that allow the filtration of biomolecules are available for size-exclusion-based separation methods in electrochemical and optical biosensing[73,118]. In addition to such methods, the concept of polymeric porous nanofilters on the FET biosensors is

introduced here as a strategy for physically and chemically structured electrical interfaces, which allow the reduction in the intensity of nonspecific electrical signals from small biomolecules as well as biomacromolecules as interfering species, which results in the increase in S/N[70-72].

## Polymeric nanofilter-based structural interfaces

An Au electrode, which has a strong catalytic action, enhances the oxidation of organic compounds, including small biomarkers such as glucose[119-121]. Also, the Au surface should be oxidized upon exposure to UV/ozone[122], so that the oxidized surface is easily reduced through the redox reaction at the surface with oxidizable biomolecules. Such Au-thin film is often utilized as the gate electrode of an extended-Au-gate–FET (EG-Au–FET), in which the gate electrode is extended from the metal gate of FET and can be handled separately from the FET. This is why not only are probe molecules easily modified on the Au gate electrode by −S–Au binding and so forth, but the Au electrode also enables the highly sensitive detection of small biomolecules owing to the redox reaction but without detection specificity or selectivity. Therefore, it is important to impart the selectivity for target detections on the Au surface; that is, only a small target biomolecule reaches and interacts with the Au surface, whereas interfering species should be filtered by some membranes on the Au surface. Here, a polymeric nanofilter interface is designed and synthesized on the Au gate electrode of an EG-Au–FET in accordance with this strategy, as shown in Fig. 4a[70-72]. The polymeric nanofilter interface, that is, the physically and chemically structured electrical interface, is composed of two layers: an anchor layer and a filter layer. The anchor layer is coated on the Au gate surface to be thinner under the filter layer, which determines the gap of the sensing capability at the Au substrate (X) and the density of the polymeric nanofilter (Y). The filter layer has the chromatographic effect, in which small interfering species are trapped but through which a small target biomolecule can reach the Au

gate surface; that is, a higher S/N is obtained for the detection of a small target biomolecule. Note that potentiometric biosensors such as the FET biosensors detect changes in the density of ionic and molecular charges within the diffusion layer of a few nanometers on the detection surface, considering the Debye length limitation[18,29,60–68]. This is why the filter layer should be located at a distance larger than the Debye length. Thus, non-specific signals generated in the filter layer are not physicochemically output as actual signals.

Surface modification by electrografting aryl diazonium salts on electrodes such as Au has attracted considerable attention because of the high chemical stability, the ease of functionalization, and the flexibility of surface design[123–125]. Aryl diazonium salts are reduced to the corresponding radicals at a reductive potential applied at the Au electrode. The generated radicals covalently react with the Au electrode surface to form a stable film, which tolerates UV irradiation[126]. Moreover, a multilayered aryl film is formed by chain reactions between unreacted radicals and benzene rings at the aryl film grafted on the Au electrode, the thickness of which can be controlled to be a few nanometers,[70,71,125]. The multilayered aryl film provides significant advantages in arranging the filter layer, contributing to the controls of not only the anchor layer thickness to be about the same as the Debye length but also the appropriate gap between grafted aryl molecules that allows a small target biomolecule to reach the Au surface. Also, the aryl films with different functionalities can be designed by modifying aryl diazonium salts with various functional groups. As an example, the aryl diazonium salt, which has a hydroxyl group at the para-substituent of the benzene ring, was utilized to develop a multilayered film for the polymeric nanofilter interface[70]. In this case, for polymerizing the filter layer, the surface was further functionalized by modifying the initiator of SI-ATRP via esterification reactions, although even only a multilayered aryl film grafted via aryl diazonium salt chemistry sufficiently suppressed the nonspecific electrical signals generated by the adhesion of large and small biomolecules to the Au surface[71].

As shown in Fig. 4b, a polymeric nanofilter interface as a model was proposed to specifically detect L-cysteine as a small target on the Au surface and to effectively trap L-DOPA as a small interfering species in the filter layer[70]. In particular, PBA was incorporated in the filter layer to capture L-DOPA but not L-cysteine. PBA forms stable esters with diol compounds such as L-DOPA at equilibrium, as described in the above section. The binding affinity of catechol to PBA is 830 M$^{-1}$ larger than that of other diol compounds at pH 7.4[78]; thus, L-DOPA has a high affinity to PBA at pH 7.4. That is, L-DOPA is trapped in the filter layer with PBA, whereas L-cysteine reaches the Au surface. In this case, the potential may change owing to the anionic form of PBA–L-DOPA complex. However, the filter layer is located outside the detection range (i.e., the Debye length) of the EG-Au–FET sensor; thus, such unexpected noise components are not detected. Here, two types of polymeric nanofilters without and with PBA were designed to investigate the filtration ability of the polymeric nanofilter. First, an MAA-based polymer (control nanofilter) was designed, and then PBA was incorporated into the filter via amidation reactions (PBA-based nanofilter). In this model, a photo mediated SI-ATRP method was employed to control the thickness and composition because it was more tolerant to MAA polymerization than the conventional ATRP method[127]. In addition, HEMA was copolymerized to enhance the hydrophilicity of the filter layer, so that L-cysteine reached the Au electrode surface through the nanofilter as expected, whereas the nonspecific adsorption of biomacromolecules such as interfering proteins (e.g., albumin) was effectively suppressed. Lastly, the filter layer was cross-linked to form a rigid polymer structure using the hydrophilic cross-linker MBA. This is because the absence of cross-linking would generate electrical noise arising from changes in the conformation of the polymeric nanofilter induced by the binding of L-DOPA with PBA. Actually, the nonspecific signals based on L-DOPA were effectively reduced by ~70–90% (□ → Δ), whereas L-cysteine was specifically detected at nM levels (◇) using the polymeric nanofilter-coated EG-Au-FET with PBA, as shown in Fig. 4c[70]. Thus, L-DOPA was successfully trapped by the nano filter

outside the detection range of the EG-Au–FET; therefore, the polymeric nanofilter contributed to the increase in S/N for the detection of L-cysteine. Additionally, the polymeric nanofilter with hydrophilicity structurally excluded nonspecific noise signals from the interaction between a few biomacromolecules such as albumin and the Au electrode[70–72].

Considering the above concept of a structural polymeric nanofilter interface, the relatively thicker MIPs mentioned in the above section may also function as the nanofilter at a distance from the gate surface above the Debye length. Moreover, they supply electrical charges based on the interaction with target biomolecules to the FET biosensor, which is captured in the cavities in the vicinity of the gate surface. This means that the thinner MIP film grafted by SI-ATRP on the gate surface should effectively contribute to the generation of electrical signals of the FET biosensor based on target biomolecules without any loss of undetected signals captured in it.

## Aptamer nanofilter-based structural interfaces

Aptamers are also candidates for a polymeric nanofilter. For instance, DA and L-DOPA were discriminated form the electrical signals with the aptamer nanofilter-modified FET biosensor despite their similar chemical structures (Fig. 4d)[55]. That is, PBA, which has been described in the above section, cannot be used as a receptor in the nanofilter because it binds to catechol amines such as DA and L-DOPA. This is why an aptamer molecule such as deoxyribonucleic acid (DNA), ribonucleic acid (RNA), or a peptide can be used as one of the structural nanofilter interfaces, although it is mostly utilized as a receptor to detect a target. Some aptamers with different base sequences for DA recognition have been reported (e.g., 44 mer)[54], whereas the base sequence of the L-DOPA aptamer is probably undetermined. The size of aptamer molecules can be relatively large (>a few nm), although it depends on the base length. Therefore, the DA aptamer can be modified on an anchor layer as the nanofilter to capture DA molecules in the undetectable region, where the height exceeds the Debye length on the EG-Au–FET[70,124,125]. Here, 1× PBS (pH 7.4) was employed as the measurement solution to demonstrate the concept of aptamer nanofilter as the chemically and physically structured interface[55], for which the Debye length is calculated to be less than 1 nm. Actually, using the EG-Au–FET, the DA aptamer layer served as the nanofilter interface to prevent DA molecules from coming in contact with the sensing surface, and L-DOPA molecules passed through the DA aptamer nanofilter and then were specifically detected by the redox reactions with the sensing surface (Fig. 4d). Thus, the structural aptamer nanofilter interface led to the increase in S/N for the detection of L-DOPA.

On the other hand, aptamers are utilized as a receptors on electrode surfaces to specifically and selectively trap and detect target biomolecules by overcoming the Debye length limitation with the FET biosensors[54]. For instance, the negatively charged backbones of aptamers with a stem–loop structure are able to approach the gate surface owing to structural reorientation based on the selective binding with a target biomolecule; that is, the negative charges of aptamers enter the diffusion layer (i.e., the Debye length) that is less affected by counterions, resulting in the generation of electrical signals of the FET biosensors. Whether aptamers are utilized as a receptor to trap and detect target biomolecules at the electrode surface or to trap but disable interfering species in the nanofilters, we need to consider the Debye length limitation. That is, not only should the ionic strength in the measurement solutions be controlled to be lower or higher, but the thickness of the anchor layer should also be designed to be thinner or thicker.

In this section, the chemically and physically structured interfaces have been introduced to decrease nonspecific noise signals, that is, increase S/N in the FET biosensors. In general, hydrophilic polymers are coated to suppress nonspecific adsorptions of large interfering species such as proteins on the gate electrode. On the other hand, filtering membranes such as polymeric nanofilters can prevent even small interfering species from approaching the gate electrode surface. Increasing S/N is very important for the FET biosensors that directly detect targets with charges in real samples, including various interfering species with charges.

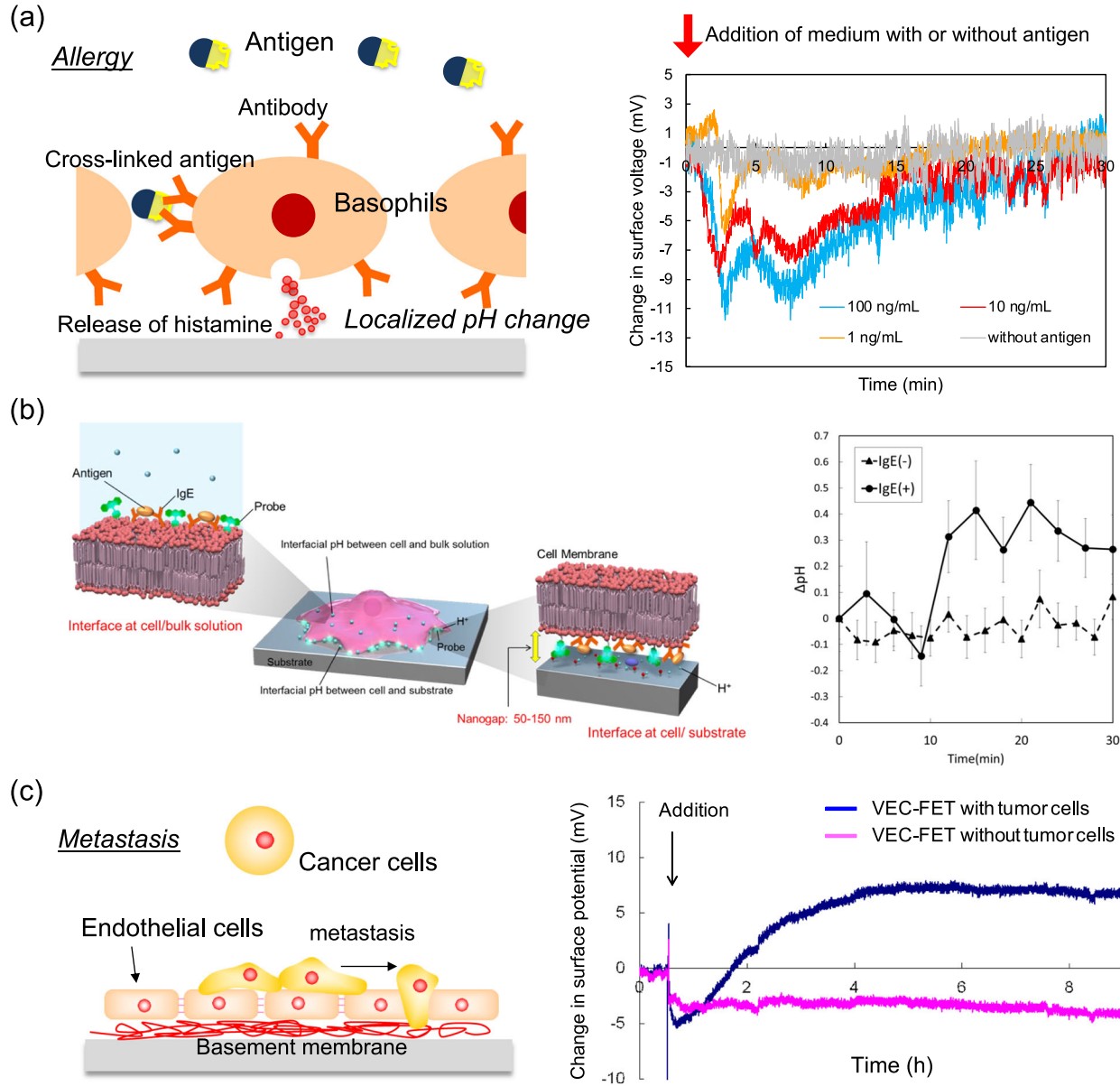

**Fig. 5 | Cellular layer as biologically induced interface. a** RBL-2H3-cell-based FET. RBL-2H3 cells were cultured on the gate oxide surface. Change in surface voltage was monitored in real-time using IgE-bound RBL-2H3-cell-based FET. Cell culture medium with or without antigen was added in the chamber, including the IgE-bound RBL-2H3-cell-based FET. The antigen concentration was controlled to 1, 10, or 100 ng/mL in the measurement solution (culture medium). Credit: From ref. 56. Reprinted with permission from the American Chemical Society. **b** Schematic illustration of RBL-2H3 cell on the substrate (left). DHPE was utilized as the extracellular pH probe. The z axis was set in the normal direction to a glass substrate. Two regions around a cell were focused on for fluorescence observation: the cell/substrate interface and the cell/bulk solution interface. A nanogap of ~50–150 nm is assumed to be at the cell/substrate interface. Change in interfacial pH at the interface between the mast cell and substrate for incubation time (right). IgE-bound [IgE(+)] or unmodified [IgE(−)] RBL-2H3 cells were used for reaction with the antigen (50 ng/mL). Interfacial pH was analyzed on the basis of the ratio of fluorescence intensity to the calibration curve. The data presented are the average of five cells with IgE and seven cells without IgE. Credit: From ref. 57. Reprinted with permission from the American Chemical Society. **c** Schematic illustration of vascular endothelial-cell-based FET (left). Change in surface potential of endothelial-cell-based FET measured for analysis of invasion of cancer cells (right). HeLa cells were added onto the endothelial-cell-based FET biosensor for in situ monitoring of invasive cancer cells, whereas the culture medium without HeLa cells was added onto the endothelial-cell-based FET as the control sensor. Credit: From ref. 59. Reprinted with permission from Wiley.

## Biologically induced cellular electrical interfaces for FET-based biosensing

When living cells are confluently cultured on an electrode surface, the cell layer itself can serve as a biologically induced cellular electrical interface to detect analytes (Fig. 5). That is, target biomolecules specifically and selectively react with cell membrane proteins such as antibodies or are taken into cells through membrane proteins such as transporters. Then, such cells are mostly activated, depending on each cell function. Here, we need to consider what is transduced into electrical signals from such activated cells. As an example, rat basophilic leukemia (RBL-2H3) cells were cultured as a signal transduction interface to induce immunological reactions on the gate oxide surface of a FET biosensor (Fig. 5a)[56,57]. This is because IgE antibodies, which bind to Fcε receptors at the RBL-2H3 cell membrane, are specifically cross-linked with allergens (antigens), resulting in the immunological response of RBL-2H3 cells[128]. The specific binding of antigens to IgE is simply detected by immunological assays. However, such methods cannot be employed to determine whether the antigen-bound IgE activates basophils in patients. This means that what is more important in the diagnosis of type I allergy is to

evaluate the potential of antigen-specific IgE to activate basophils from a patient rather than the detection of IgE–antigen binding only. In fact, secretions mainly including histamine, which were released during immunological reactions based on the IgE–antigen reaction, were indirectly detected as the change in pH with the FET biosensor[56,57]. Histamine is mainly released from RBL-2H3 cells in a relatively short time through immunological reactions and shows basicity (pKa 9.75) in particular. An immunological response was observed within 10 min upon adding the antigen [dinitrophenyl-conjugated human serum albumin (DNP-HSA)] to RBL-2H3 cells with the FET biosensor; therefore, basic histamine had the greatest impact on electrical signals. This is why secreted histamines are considered to be close to the cell/gate interface, resulting in an increase in pH around it. The FET biosensor used was based on a pH-sensitive FET, the gate of which was composed of $Ta_2O_5$ with hydroxyl groups in a solution[1,5,7]. The pH-sensitive FETs used showed a response of 55.8 mV/pH, which was near the Nernstian response. Here, RBL-2H3 cells were cultured on the gate oxide surface, so we need to consider a gap of approximately 50–150 nm between the cells and the gate oxide. The pH behavior at the cell/substrate nanogap was observed by laser scanning confocal microscopy using phospholipid fluorescein inserted at the cell membrane (Fig. 5b)[25,57]. In addition, the cell culture gate FET biosensors continuously monitored the change in pH on the basis of cellular respiration (metabolism) at the cell/gate nanogap interface for a few days to one month[24,31,32,129,130]. Some proteins contained in a cell culture medium are nonspecifically adsorbed at the gate oxide surface during preculture, contributing to the adhesion of cells at the gate oxide surface. These biomacromolecules prevent target biomolecules with charges from approaching the gate oxide surface, but small hydrogen ions easily pass through such biomacromolecules and then react hydroxy groups at the gate oxide surface. This means that the change in pH at the cell/gate nanogap interface contributes to the change in $V_{out}$ according to the principle of FET biosensors. In addition, hydrogen ions should be effectively concentrated in the closed nanogap space between the cell membrane and the gate surface, which can be easily detected. This is the simplest way to use pH-sensitive FETs[129].

As another example of biologically induced cellular electrical interfaces, vascular endothelial cells were confluently cultured as a cell layer on the gate electrode surface of a FET biosensor to monitor the invasion of cancer cells (Fig. 5c)[59]. Metastatic cancer cells adhere to vascular endothelial cells and then invade into surrounding tissues through basement membrane, which results in metastasis[58]. Analytical methods to monitor metastasis in real time contribute to not only the early diagnosis of cancer but also the clarification of the mechanism of metastasis. Here, vascular endothelial cells were utilized as a biologically induced interface material on the FET biosensor to simulate the interaction with cancer cells in blood. Actually, the invasion process of cancer cells (HeLa cells) into the vascular endothelial cell [human coronary artery endothelial (HCAE) cell] layer was continuously monitored in the cell culture medium (pH 7.4) using the FET biosensor[59]. Note that the basement membrane, which is a thin and fibrous extracellular matrix of tissue composed of collagen IV and so forth and negatively charged at pH 7.4, should be formed on the gate oxide surface by HCAE cells[131]. The basement membrane is also known to be decomposed by collagenase secreted by cancer cells during the invasion process[132]. As a result, $V_{out}$ gradually increased after adding HeLa cells onto the HCAE-cell–FET, which resulted from the change in the density of some kind of electrical charge on the gate electrode, but $V_{out}$ for the HCAE-cell–FET without HeLa cells hardly changed because only the culture medium was added (right graph in Fig. 5c)[59]. From the change in the fluorescence image of collagen IV in the basement membrane, furthermore, collagen IV was observed under the HCAE cell layer before adding Hela cells, whereas it was partially decomposed after introducing HeLa cells onto HCAE cells. Thus, HeLa cells are considered to have passed between HCAE cells and then decomposed the basement membrane by secreting a collagen-IV-decomposing enzyme. Considering the electrical signal measurement and fluorescence observation results, $V_{out}$ must have increased by the decomposition of the negatively charged basement membrane, that is, the invasion

of cancer cells, corresponding to the decrease in the density of negative charges. Thus, vascular endothelial cells with the basement membrane serve as the biologically induced cellular electrical interface that provides the specificity and $S$ for the detection of cancer cells as a target and induces the generation of electrical signals.

In this section, we have introduced that unique cellular functions contribute to the specificity to and $S$ for targets detected. Such biologically derived functional materials are useful as the biologically induced interfaces in the FET biosensors, including enzymes and antibodies.

## Conclusion and outlook

In developing a biosensor, we consider the design criteria based on its three components, namely, the biological target, signal transduction interface, and detection device. Among the detection devices, a platform based on an electronic device with the FET biosensors is suitable for use in miniaturized and cost-effective systems to directly measure biological samples because the FET biosensors allow the direct detection of intrinsic ionic and biomolecular charges in principle, which contributes to label- and enzyme-free biosensing. Such miniaturized electronic devices can be easily equipped with a wireless function and attached to the body, which is available for wearable biosensors to detect biomarkers in a blood-sampling-free manner (i.e., tears, sweat, and saliva). Here, it is very important to determine how the change in the density of charges based on biomolecular recognition events is directly transduced into electrical signals at the signal transduction interfaces, regardless of the wearability of FET biosensors. Such bio/device interfaces are chemically synthesized, physically and chemically structured, and biologically induced to control the biosensing parameters such as specificity, $S$, $K_a$, LOD, S/N, and biocompatibility with respect to the biological target, although the chemically synthesized electrical interfaces are also useful as the signal transduction interfaces for the flexible wearable biosensors. In particular, the increase in $K_a$ for the target biological target, which results in the enhancement of $S$ and LOD, becomes a key challenge for enzyme-free interfaces, and then biocompatible materials may be chosen for the signal transduction interfaces. On the other hand, various semiconductor materials have been recently applied as the channel of FET biosensors. Therefore, the diversity of signal transduction interfaces broadens the possibility of developing novel biosensing devices, in parallel with the development of new channel materials for the FET biosensors. In addition, the steep SS, which is one of the transistor characteristics, should lead to an increase in the sensitivity of biosensing, although it is a key parameter of FETs (see 1. Introduction)[27,31], and then the optimal biosensing parameters are effectively provided by the diverse signal transduction interfaces for the practical use of FET biosensors.

In addition, the FET biosensors can be applied to semiconductor integrated circuits to measure multiple samples simultaneously. This is one of the advantages of utilizing semiconductor technology and also the unique feature of FETs because other biosensors (e.g., surface plasmon resonance and quartz crystal microbalance sensors) hardly enable the integration of electrodes as in complementary metal oxide semiconductor sensors[21]. That is, it will be a challenge to coat and arrange different signal transduction interface materials for various biomarkers on the individual gate electrodes in the arrayed devices in the future. This means that the methods to analyze a huge amount of data, including complicated information, must be required, such as the omics approaches.

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

## Acknowledgements

We would like to thank all the members of Sakata Laboratory for their help and useful discussion.

## Competing interests

The author declares no competing interests.
