## [Peer Review File · Communications Chemistry]

Reviewers' comments:

Reviewer #1 (Remarks to the Author):

This manuscript systematically describes the diverse signal transduction interfaces, such as chemically synthesized, physically and chemically structured, and biologically induced interfaces, on the gate electrode of field-effect transistor (FET) biosensors. The impacts of the diverse signal transduction interfaces on some biosensing parameters (e.g., specificity, selectivity, signal-to-noise ratio, and the limit of detection) and characteristics (e.g., biocompatibility) of sensors are introduced. However, it should be pointed out that the summary of literature in this review is not comprehensive, and some important literature has not been cited (e.g., Nat. Commun. 2020, 11, 3226; J. Am. Chem. Soc. 2009, 131, 4788-4794.; J. Am. Chem. Soc., 2009, 131, 12022-12023.; Chem. Rev. 2008, 108, 826-844.; Anal. Chem. 1977, 49, 2315-2321.). Additionally, potentiometric field-effect transistors (FET) cannot represent potentiometric sensors. In recent years, ion selective electrodes and light-addressable potentiometric sensors, as the important parts of potentiometric sensors, have also experienced prosperous development. The scope of the title of the paper does not match the content of the main text. Therefore, in my opinion, if the manuscript wants to be published in Communications Chemistry, the above issues need to be addressed.

Reviewer #2 (Remarks to the Author):

This review highlights the importance of signal transduction interfaces in electrochemical biosensors, particularly enzyme-free biosensors. It introduces various interfaces, including the potentiometric field-effect transistor (FET) biosensor, which enables direct detection of charges for label-free and enzyme-free biosensing. The diverse signal transduction interfaces on the gate electrode of FET biosensors plays a key role in controlling biosensing parameters for biological targets and wearable biosensors. However, before considering further publication, there are some suggested revisions.

1. The importance and urgency of this article should be emphasized, including whether similar reviews have been conducted in this field before.
2. It is advisable to use a limited number of individual article citations, avoiding instances such as excessive referencing to specific figures like "Figure 2a" within the manuscript.
3. The article frequently mentions wearable sensors, and it is crucial to provide a more detailed explanation of the importance of signal transduction interfaces in relation to these applications.
4. It is recommended to include some important related work about biosensors, such as Biosens. Bioelectron. 2020, 148, 111799; Chem. Soc. Rev. 2020, 49 (13), 4405-4465.; Biosens. Bioelectron. 2020, 170: 112636.
5. The article primarily focuses on FET and would benefit from providing a foundational description of FET, including its principles, advancements, and its advantages compared to other detection methods.
6. If functional polymers mentioned in the conclusion are not discussed in the main body of the review, readers may expect them to be incorporated into the main text rather than being introduced in the conclusion.
7. The conclusion and discussion of the review could benefit from more in-depth insights, such as addressing the performance advancements in miniaturized FETs and providing guidance on details that readers may be interested in, including considerations and directions for future development.
8. In addition to the comparative listing of advancements in MIPs, it is equally important to provide similar discussions on other vertical sensing modalities.
9. In addition to emphasizing their significance, the figures in the review should be visually appealing, avoiding excessive white space and the use of overly vibrant colors.
10. The overall content of the article is relatively limited, with some discussions being excessively broad and shallow. In addition to literature references, the author should provide more original insights and prospects for the development of the field. More constructive guidance and recommendations on limitations and challenges would be highly valuable.

Reviewer #3 (Remarks to the Author):

In this manuscript, authors aim to highlight recent advances in the electrochemical interfaces which are one of the key issues for biosensors. In particular, the diverse signal transduction interfaces are summarized for enzyme-free electrochemical biosensor for example, chemically synthesized, physically or chemically structured interfaces. These features are important for controlling biosensing performance such as selectivity, sensitivity, limit of detection or S/N ratio). Considering these issues, this paper is interesting in this area, and is well organized and beneficial for the researchers or readers of this journal. However, there are some weaknesses to be addressed before publication. I recommend the major revision for this manuscript. My questions are as follows:

1. It is suggested to add more paragraph with respect to the field-effect transistor in the introduction.
2. It is suggested to modify the Abstract to highlight the innovation of this paper.
3. The advantage/disadvantages in each section (from 2.1 to 4) should be briefly discussed in the text.
4. The more advanced functional interfaces, its detailed description of the sensing mechanism, advantage and disadvantages should be clearly presented. In particular, the key challenges are not presented.
5. It is recommended that the authors should add more references along with the detailed sensing performances such as sensitivity, S/N ratio, limit of detection in Table 1, not just focusing in only MIP-based sensor. In addition, if applicable, the advantages and disadvantages of each biosensor would be good in the Table 1.
6. It would be better to the readers when author could change the Figure 1 with whole schematic diagram showing different interface, characteristics and their corresponding detection method for the target detection.

7. The outlook viewpoints or key challenges should be presented in the conclusion section. I would recommend to re-write this conclusion section.
8. Figure 2 should be changed because this is little confusing.
9. The section 3.2 in the text should be re-written.
10. It would be better that the comparison of 3 different interfaces described in the text (advantage/disadvantage, insight or key issue) is presented as another Table.
11. There are some typographical errors in the reference section. It means that the first word should start with a capital letter. Please address accordingly.

Dear Editor

Thank you very much for your kind reviews. We have revised our manuscript in accordance with the reviewers' comments as follows. We would be grateful if the manuscript could be reevaluated for publication in *Communications Chemistry* [Review Article].

[Reviewer #1]

Comments:

This manuscript systematically describes the diverse signal transduction interfaces, such as chemically synthesized, physically and chemically structured, and biologically induced interfaces, on the gate electrode of field-effect transistor (FET) biosensors. The impacts of the diverse signal transduction interfaces on some biosensing parameters (e.g., specificity, selectivity, signal-to-noise ratio, and the limit of detection) and characteristics (e.g., biocompatibility) of sensors are introduced. However, it should be pointed out that the summary of literature in this review is not comprehensive, and some important literature has not been cited (e.g., Nat. Commun. 2020, 11, 3226; J. Am. Chem. Soc. 2009, 131, 4788-4794.; J. Am. Chem. Soc., 2009, 131, 12022-12023.; Chem. Rev. 2008, 108, 826-844.; Anal. Chem. 1977, 49, 2315-2321.) → Reply 1. Additionally, potentiometric field-effect transistors (FET) cannot represent potentiometric sensors. In recent years, ion selective electrodes and light-addressable potentiometric sensors, as the important parts of potentiometric sensors, have also experienced prosperous development. The scope of the title of the paper does not match the content of the main text. → Reply 2 Therefore, in my opinion, if the manuscript wants to be published in *Communications Chemistry*, the above issues need to be addressed.

(Reply 1)

Thank you very much for your kind review. In accordance with the reviewer's comment, we have added references (6) and (14) in the revised manuscript. Then, we have added the following sentences with their references cited and **Figure 1b**.

“(6) Buck, R. P. & Hackleman, D. E. Field Effect Potentiometric Sensors. *Anal. Chem.* **49**, 2315–2321 (1977).”

“(14) Fakhri, I., Durnan, O., Mahvash, F., Napal, I., Centeno, A., Zurutuza, A., Yargeau, V. & Szkopek, T. Selective ion sensing with high resolution large area graphene field effect transistor arrays. *Nat. Commun.* **11**, 3226 (2020).”

“A platform based on a solution-gated field-effect transistor (FET), which originates from electronics, is suitable for use in miniaturized and cost-effective systems to directly

measure biological samples as the FET biosensor in the field of *in vitro* diagnostics.¹ Such miniaturized electronic devices can be easily equipped with a wireless function and attached to the body, which are available for wearable biosensors to detect biomarkers in tears, sweat, and saliva, that is, for diagnostics in a blood-sampling free manner.^{2–4} In general, the gate insulator surface (e.g., SiO₂) is directly in contact with a measurement solution in the FET biosensor without a metal gate electrode, which is different from a metal-oxide-semiconductor (MOS) transistor, for which the potential of the measurement solution is controlled by the reference electrode,^{5–7} as shown in **Figure 1a**. When ions or biomolecules with charges are adsorbed on the gate insulator surface, their charges electrostatically interact with electrons across the gate insulator, resulting in a change in the conductivity of the channel of the FET. That is, the drain–source current (I_{DS}) at the channel changes with the change in the density of ions or biomolecules with charges adsorbed on the gate insulator. That is, such charged species induce a change in the interfacial potential (V_{out}) of the solution/gate insulator at a constant I_{DS} , which is potentiometrically detected.^{5,8} Oxide and nitride membranes used as the gate insulator and the passivation layer can effectively detect a change in pH on the basis of the reaction in equilibrium between hydrogen ions and hydroxy groups at their surfaces.⁵ Afterwards, various ion-sensitive membranes (ISMs)^{2,9–15} and biomimetic receptors with enzymes, antibodies, and single-stranded DNAs^{16–21} were coated on the gate electrode surfaces as the FET biosensors to specifically and selectively detect target ions and biomolecules, and further cellular activities were monitored on the basis of the change in ionic behaviors at the cell/gate interface in real time.^{22–25} This is because the detection principle of FET biosensors is basically derived from the potentiometric measurement of the changes in ionic and biomolecular charges or membrane capacitances at the electrolyte solution/gate electrode interface. Moreover, various semiconductive materials have been widely utilized as the channel of FETs for biosensing devices, such as one-dimensional [1D (e.g., nanotubes and nanowires)] and two-dimensional [2D (e.g., graphene, diamond, and MoS₂)] materials.^{15,26–30} In particular, a solution-gated 1D or 2D-channel FET biosensor, the channel of which is directly in contact with an electrolyte solution, is expected to have a steep subthreshold slope (SS), resulting in an ultrahighly sensitive biosensing, owing to a relatively large capacitance of the electric double-layer at the electrolyte solution/channel interface.^{27,31} Moreover, thin-film transistors (TFTs) such as transparent amorphous oxide semiconductors (e.g., amorphous In–Ga–Zn–oxide) can be applied as one of the FET biosensors, which are deposited on transparent substrates such as glass and plastics.³² Thus, the number of research studies on the FET biosensors is increasing yearly (**Figure 1b**).” (see pages 3 and 4 in the revised and highlighted manuscript)

(Reply 2)

Thank you very much for your kind review. In accordance with the reviewer's comment, we have reconsidered and modified the title; that is, we have focused on field-effect transistor-based biosensors with diverse signal transduction interfaces rather than discussing "potentiometric methods" in general. This means that we may not need to provide specific comments about light-addressable potentiometric sensors. We would be grateful if the reviewer could understand our concept in this review article.

"Diversity of signal transduction interfaces for field-effect transistor-based biosensors"

*In addition, we have thoroughly modified the manuscript in accordance with the comments from other reviewers.

[Reviewer #2]

Comments:

This review highlights the importance of signal transduction interfaces in electrochemical biosensors, particularly enzyme-free biosensors. It introduces various interfaces, including the potentiometric field-effect transistor (FET) biosensor, which enables direct detection of charges for label-free and enzyme-free biosensing. The diverse signal transduction interfaces on the gate electrode of FET biosensors plays a key role in controlling biosensing parameters for biological targets and wearable biosensors. However, before considering further publication, there are some suggested revisions.

1. The importance and urgency of this article should be emphasized, including whether similar reviews have been conducted in this field before.

(Reply)

Thank you very much for pointing this out. In accordance with the reviewer's comment, we have revised the manuscript to show the importance and urgency of this article (see pages 2 to 4, 6, and 7 for Abstract and Introduction and pages 27 and 28 for Conclusion and outlook in the revised and highlighted manuscript). In particular, we have added the following sentences and **Figure 1b** in the revised manuscript to highlight the importance and urgency of this article.

Abstract:

“A platform of a field-effect transistor (FET) biosensor based on electronics is suitable for use in miniaturized and cost-effective systems that are required in the field of *in vitro* diagnostics. This enables the direct detection of ionic or biomolecular charges in a biosample, which contributes to label-free biosensing. In addition, various semiconductive materials have been applied as the channel of FETs for biosensing, such as one- and two-dimensional materials. Thus, studies on FET biosensors are intensively pursued, the number of which is rapidly increasing. Here, a signal transduction interface material between the biosample and the channel of FETs plays a key role in capturing target ions or biomolecules, which then induces their electrochemical reactions into output signals. A versatile concept for a signal transduction interface is required for various biomarkers because there are no enzymes or antibodies applicable to every target biomarker. In this review, distinctive signal transduction interfaces for the FET biosensors are introduced, such as chemically synthesized, physically structured, and biologically induced interfaces, without relying on enzymes or antibodies. Diverse signal transduction interfaces in the FET biosensors become a key element in controlling biosensing parameters, such as specificity, selectivity, binding constant, limit of detection, signal-to-noise ratio, and biocompatibility.” (see page 2 in the revised and highlighted manuscript)

In 1. Introduction

“Moreover, various semiconductive materials have been widely utilized as the channel of FETs for biosensing devices, such as one-dimensional [1D (e.g., nanotubes and nanowires)] and two-dimensional [2D (e.g., graphene, diamond, and MoS₂)] materials.^{15,26–30} In particular, a solution-gated 1D or 2D-channel FET biosensor, the channel of which is directly in contact with an electrolyte solution, is expected to have a steep subthreshold slope (SS), resulting in an ultrahighly sensitive biosensing, owing to a relatively large capacitance of the electric double-layer at the electrolyte solution/channel interface.^{27,31} Moreover, thin-film transistors (TFTs) such as transparent amorphous oxide semiconductors (e.g., amorphous In–Ga–Zn–oxide) can be applied as one of the FET biosensors, which are deposited on transparent substrates such as glass and plastics.³² Thus, the number of studies on the FET biosensors is increasing yearly (**Figure 1b**).” (see page 4 in the revised and highlighted manuscript)

“Considering the application of electric devices with FETs to biosensors, it is very important to design devices such that the change in the density of charges based on biomolecular recognition events is directly transduced into electrical signals at the signal transduction interfaces without relying on enzymes and antibodies. Therefore, the signal

transduction interfaces classified as *chemically synthesized, physically and chemically structured, and biologically induced interfaces* become a key element in controlling the performances of FET biosensors, which determine their future applications. However, there are no reports on the signal transduction interfaces as classified in the above.” (see page 7 in the revised and highlighted manuscript)

“In particular, the diverse signal transduction interfaces in FET biosensors are noted to become a key element in controlling biosensing parameters with respect to biological targets, such as specificity, selectivity, K_a , LOD, S/N, and biocompatibility. Therefore, the diversity of signal transduction interfaces should broaden the possibility of developing novel biosensing devices, in parallel with the development of new channel semiconductors of FET biosensors.” (see page 7 in the revised and highlighted manuscript)

2. It is advisable to use a limited number of individual article citations, avoiding instances such as excessive referencing to specific figures like “Figure 2a” within the manuscript.

(Reply)

In accordance with the reviewer’s helpful suggestion, we have reconsidered the references throughout the manuscript; 24 references shown in the original manuscript have been deleted, and 36 references have been added in the revised manuscript.

3. The article frequently mentions wearable sensors, and it is crucial to provide a more detailed explanation of the importance of signal transduction interfaces in relation to these applications.

(Reply)

Indeed, regardless of the wearability and flexibility of biosensors, the signal transduction interfaces should be considered to control the biosensing parameters for their applications. In particular, a platform based on a solution-gated field-effect transistor (FET), which originates from electronics, is suitable for use in miniaturized and cost-effective systems to directly measure biological samples as the FET biosensor in the field of *in vitro* diagnostics.¹ Such miniaturized electronic devices can be easily equipped with a wireless function and attached to the body, which are available for wearable biosensors to detect biomarkers in tears, sweat, and saliva, that is, for diagnostics in a blood-sampling free manner. Moreover, it is better for the wearable biosensors to be flexible. In this case, it is better to use soft materials such as polymeric membranes or a thin film as the signal transduction interfaces to decrease stiffness. In that regard, the chemically synthesized

electrical interfaces are useful as the signal transduction interfaces for flexible wearable biosensors. Therefore, we have added the following sentences in the revised manuscript.

“A platform based on a solution-gated field-effect transistor (FET), which originates from electronics, is suitable for use in miniaturized and cost-effective systems to directly measure biological samples as the FET biosensor in the field of *in vitro* diagnostics.¹ Such miniaturized electronic devices can be easily equipped with a wireless function and attached to the body, which are available for wearable biosensors to detect biomarkers in tears, sweat, and saliva, that is, for diagnostics in a blood-sampling free manner.²⁻⁴” (see page 3 in the revised and highlighted manuscript)

“Such flexible polymeric membranes are also available for wearable biosensors. That is, it is better to use soft materials such as polymeric membranes or a thin film as the signal transduction interfaces to decrease the stiffness of the wearable and flexible biosensors.” (see page 5 in the revised and highlighted manuscript)

“Most popular ion sensors are composed of hydrophobic and flexible polymeric membranes such as polyvinyl chloride (PVC) including ionophores coated on electrodes for the potentiometric measurements of target ions. Artificial ionophores such as crown ethers and calixarenes are dissolved in a plasticizer to enhance their mobilities in the flexible PVC membrane. Various ions such as Na⁺, K⁺, Ca²⁺, NH₄⁺, and Cl⁻ can be detected using the potentiometric ISM-coated electrodes including FETs (**Figure 2a**).^{2,9-15} Thus, flexible ISMs enable the detection of inorganic ions in biological and environmental samples.” (see page 8 in the revised and highlighted manuscript)

“Moreover, the biocompatibilities of ISMs with flexibility should be considered for wearable biosensors, focusing on cytotoxicity and biofouling.” (see page 9 in the revised and highlighted manuscript)

“Thus, the modified ISM-based FETs can be utilized more safely and precisely in actual biological samples with interfering species as wearable biosensors with flexibility.” (see page 10 in the revised and highlighted manuscript)

“An MIP is a biomimetic and soft material designed for selective molecular recognition.³⁹⁻⁴³” (see page 10 in the revised and highlighted manuscript)

“In addition, stiffer MIP matrices may be more selective; that is, it should be beneficial to prepare flexible polymeric membranes as rigid as possible.⁹⁵ Actually, electroactive polymeric membranes with relatively high Young’s moduli such as polypyrrole (PPy) and poly(*o*-phenylenediamine) (PoPDA) appear to be utilized as the MIP matrices, preserving the macromolecular arrangements after the template extraction.⁹⁶⁻¹⁰² On the other hand, 2-hydroxyethylmethacrylate (HEMA), ethylene glycol dimethacrylate

(EGDMA), and so forth are copolymerized to improve the hydrophilicity of MIP matrices with flexibility, which include small biomolecules and electrolytes and may show relatively low Young's moduli, when PBA derivatives (e.g., 4-vinyl-PBA) are copolymerized in the MIP matrices on the gate electrode (**Figure 3b**).^{44,50–53} (see page 15 in the revised and highlighted manuscript)

“although the chemically synthesized electrical interfaces are also useful as the signal transduction interfaces for the flexible wearable biosensors.” (see page 27 in the revised and highlighted manuscript)

4. It is recommended to include some important related work about biosensors, such as Biosens. Bioelectron. 2020, 148, 111799; Chem. Soc. Rev. 2020, 49 (13), 4405–4465.; Biosens. Bioelectron. 2020, 170: 112636.

(Reply)

Thank you very much for this helpful suggestion. In accordance with the reviewer's comment, we have reconsidered the references throughout the manuscript and then added the following reference.

“(15) Shao, Y., Ying, Y. & Ping, J. Recent advances in solid-contact ion-selective electrodes: functional materials, transduction mechanisms, and development trends. *Chem. Soc. Rev.* **49**, 4405–4465 (2020).”

5. The article primarily focuses on FET and would benefit from providing a foundational description of FET, including its principles, advancements, and its advantages compared to other detection methods.

(Reply)

Thank you very much for this helpful suggestion. In accordance with the reviewer's comment, we have modified Abstract and then added some introductory information on the FET biosensor in Introduction, including its principle, advantages, and so forth, as follows. In line with this revision, we have modified the references. In particular, **Figure 1b** has been added in the revised manuscript.

In Abstract:

“A platform of a field-effect transistor (FET) biosensor based on electronics is suitable for use in miniaturized and cost-effective systems that are required in the field of *in vitro* diagnostics. This enables the direct detection of ionic or biomolecular charges in a biosample, which contributes to label-free biosensing. In addition, various

semiconductive materials have been applied as the channel of FETs for biosensing, such as one- and two-dimensional materials. Thus, studies on FET biosensors are intensively pursued, the number of which is rapidly increasing. Here, a signal transduction interface material between the biosample and the channel of FETs plays a key role in capturing target ions or biomolecules, which then induces their electrochemical reactions into output signals. A versatile concept for a signal transduction interface is required for various biomarkers because there are no enzymes or antibodies applicable to every target biomarker. In this review, distinctive signal transduction interfaces for the FET biosensors are introduced, such as chemically synthesized, physically structured, and biologically induced interfaces, without relying on enzymes or antibodies. Diverse signal transduction interfaces in the FET biosensors become a key element in controlling biosensing parameters, such as specificity, selectivity, binding constant, limit of detection, signal-to-noise ratio, and biocompatibility.” (see page 2 in the revised and highlighted manuscript)

In 1. Introduction:

“A platform based on a solution-gated field-effect transistor (FET), which originates from electronics, is suitable for use in miniaturized and cost-effective systems to directly measure biological samples as the FET biosensor in the field of *in vitro* diagnostics.¹ Such miniaturized electronic devices can be easily equipped with a wireless function and attached to the body, which are available for wearable biosensors to detect biomarkers in tears, sweat, and saliva, that is, for diagnostics in a blood-sampling free manner.²⁻⁴ In general, the gate insulator surface (e.g., SiO₂) is directly in contact with a measurement solution in the FET biosensor without a metal gate electrode, which is different from a metal-oxide-semiconductor (MOS) transistor, for which the potential of the measurement solution is controlled by the reference electrode,⁵⁻⁷ as shown in **Figure 1a**. When ions or biomolecules with charges are adsorbed on the gate insulator surface, their charges electrostatically interact with electrons across the gate insulator, resulting in a change in the conductivity of the channel of the FET. That is, the drain–source current (I_{DS}) at the channel changes with the change in the density of ions or biomolecules with charges adsorbed on the gate insulator. That is, such charged species induce a change in the interfacial potential (V_{out}) of the solution/gate insulator at a constant I_{DS} , which is potentiometrically detected.^{5,8} Oxide and nitride membranes used as the gate insulator and the passivation layer can effectively detect a change in pH on the basis of the reaction in equilibrium between hydrogen ions and hydroxy groups at their surfaces.⁵ Afterwards, various ion-sensitive membranes (ISMs)^{2,9-15} and biomimetic receptors with enzymes,

antibodies, and single-stranded DNAs^{16–21} were coated on the gate electrode surfaces as the FET biosensors to specifically and selectively detect target ions and biomolecules, and further cellular activities were monitored on the basis of the change in ionic behaviors at the cell/gate interface in real time.^{22–25} This is because the detection principle of FET biosensors is basically derived from the potentiometric measurement of the changes in ionic and biomolecular charges or membrane capacitances at the electrolyte solution/gate electrode interface. Moreover, various semiconductive materials have been widely utilized as the channel of FETs for biosensing devices, such as one-dimensional [1D (e.g., nanotubes and nanowires)] and two-dimensional [2D (e.g., graphene, diamond, and MoS₂)] materials.^{15,26–30} In particular, a solution-gated 1D or 2D-channel FET biosensor, the channel of which is directly in contact with an electrolyte solution, is expected to have a steep subthreshold slope (SS), resulting in an ultrahighly sensitive biosensing, owing to a relatively large capacitance of the electric double-layer at the electrolyte solution/channel interface.^{27,31} Moreover, thin-film transistors (TFTs) such as transparent amorphous oxide semiconductors (e.g., amorphous In–Ga–Zn–oxide) can be applied as one of the FET biosensors, which are deposited on transparent substrates such as glass and plastics.³² Thus, the number of research studies on the FET biosensors is increasing yearly (**Figure 1b**).” (see pages 3 and 4 in the revised and highlighted manuscript)

6. If functional polymers mentioned in the conclusion are not discussed in the main body of the review, readers may expect them to be incorporated into the main text rather than being introduced in the conclusion.

(Reply)

Thank you very much for this helpful suggestion. Since functional polymers mentioned in the conclusion may mislead the readers, considering the concept of this article, we have thoroughly reconsidered the manuscript and then revised “Conclusion and outlook”, as follows.

In Conclusion and outlook:

“In developing a biosensor, we consider the design criteria based on its three components, namely, the biological target, signal transduction interface, and detection device. Among the detection devices, a platform based on an electronic device with the FET biosensors is suitable for use in miniaturized and cost-effective systems to directly measure biological samples because the FET biosensors enable the direct detection of intrinsic ionic and biomolecular charges in principle, which contributes to label- and enzyme-free biosensing. Such miniaturized electronic devices can be easily equipped with a wireless

function and attached to the body, which are available for wearable biosensors to detect biomarkers in a blood-sampling free manner (i.e., tears, sweat, and saliva). Here, it is very important to determine how the change in the density of charges based on biomolecular recognition events is directly transduced into electrical signals at the signal transduction interface, regardless of the wearability of FET biosensors. Such bio/device interfaces are chemically synthesized, physically and chemically structured, and biologically induced to control the biosensing parameters such as specificity, S , K_a , LOD, S/N, and biocompatibility with respect to the biological target, although the chemically synthesized electrical interfaces are also useful as the signal transduction interfaces for the wearable biosensors with flexibility. In particular, the increase in K_a for the target biological target, which results in the enhancement of S and LOD, becomes a key challenge for enzyme-free interfaces, and then biocompatible materials may be chosen for the signal transduction interfaces. On the other hand, various semiconductor materials have been recently applied as the channel of FET biosensors. Therefore, the diversity of signal transduction interfaces broadens the possibility of developing novel biosensing devices, in parallel with the development of new channel materials of the FET biosensors. In addition, the steep SS, which is one of the transistor characteristics, should lead to the increase in the sensitivity of biosensing, although it is a key parameter of FETs (*see* 1. Introduction),^{27,31} and then the optimal biosensing parameters are effectively provided by the diverse signal transduction interfaces for the practical use of FET biosensors.” (see pages 27 and 28 in the revised and highlighted manuscript)

7. The conclusion and discussion of the review could benefit from more in-depth insights, such as addressing the performance advancements in miniaturized FETs and providing guidance on details that readers may be interested in, including considerations and directions for future development.

(Reply)

Thank you very much for this helpful suggestion. we have revised “Conclusion and outlook”, as shown in Reply 6. We have also added some discussions in each section, as follows. In particular, **Figures 3d, 3e, S1, and S2** have been added in the revised manuscript.

In 1. Introduction:

“The Debye length λ depends on the ionic strength of the electrolyte solution used and is expressed as $\lambda = (\epsilon_0 \epsilon_r k_B T / 2 N_A e^2 I)^{1/2}$, where I is the ionic strength of the electrolyte solution, ϵ_0 is the permittivity of free space, ϵ_r is the dielectric constant, k_B is the

Boltzmann constant, T is the absolute temperature, N_A is the Avogadro number, and e is the elementary charge. The Debye length limit is controlled by changing the ionic strength of a measurement solution, that is, diluted measurement solutions are useful for improving the detection sensitivity of the FET biosensors to charged biomolecules because of the reduction of the shielding effect by counterions.” (see page 6 in the revised and highlighted manuscript)

In 2.1. Ion-sensitive membranes and their biocompatibility:

“Moreover, the selectivity of a crown ether L for ions M_1^{n+} and M_2^{n+} is expressed as the ratio of each binding constant $K_a(1) / K_a(2)$; e.g., $K_a(1) = [M_1L^{n+}] / ([M_1^{n+}] + [L])$. The complexing ability, that is, K_a of crown ether with ions, is dependent on the relative sizes of the cavity of the crown ether and the ions.⁸⁰ In addition, poly- and bis (crown ether)s are favorable for the formation of 2:1 (crown ether unit/ion) sandwich-type complexes with ions, which contribute to the increase in K_a , that is, the selectivity to a specific ion.⁸¹” (see page 9 in the revised and highlighted manuscript)

“Hydrophilic polydopamine (poly-DA) modification actually enhanced the anti-adhesive properties of ISMs by increasing the surface hydrophilicity.⁸⁶ This is why a poly-DA film prevented the nonspecific adsorptions of proteins that may generate noise signals on the ISMs, while the detection sensitivity for monovalent ions did not deteriorate, keeping it near the Nernstian response. Thus, the modified ISM-based FETs can be utilized more safely and precisely in actual biological samples with interfering species as wearable biosensors with flexibility.” (see page 10 in the revised and highlighted manuscript)

In 2.2. MIP-based electrical interfaces:

“For instance, glucose molecules used as the template, which bind to PBA, are removed under acidic conditions, whereas glucose boronate esters induce negative charges in the MIP matrices under relatively basic conditions, because the pK_a of a glucose boronate ester is 6.8.⁷⁹” (see page 11 in the revised and highlighted manuscript)

“To understand the chemical basis of interactions between the MIP and the target biomolecules underlying the electrical responses of MIP-based FET biosensors, quantitative analysis is required. In general, the characteristics of the binding of a target molecule to MIP are quantified using adsorption isotherm equations, as the binding process involves the reversible adhesion of the target molecule to the target-selective membrane.⁹¹ In this way, the potentiometric analyses based on the FET biosensor can directly characterize the MIP interface without the batch rebinding process, which is often

required for MIP characterization. According to a previous study,⁹¹ the Langmuir adsorption isotherm equation for a bulk rebinding system is expressed as

$$B = \frac{N[c]}{1 + K_a[c]}, \quad (1)$$

where B refers to a signal observed at equilibrium for the MIP-bound template, $[c]$ to the free concentration of the template at equilibrium, N to the number of available active centers in the MIP per unit volume, and K_a to the binding constant. Equation 1 assumes homogeneously distributed binding sites with a constant binding constant K_a .

The operation of a silicon-based FET in the unsaturated region can generally be described as

$$I_{DS} = \mu C_{OX} \frac{W}{L} \left[(V_{GS} - V_T) V_{DS} - \frac{1}{2} V_{DS}^2 \right], \quad (2)$$

where μ is the electron mobility in the channel, C_{OX} is the gate oxide capacitance, $\frac{W}{L}$ is the channel width-to-length ratio, V_{DS} and V_{GS} are the applied drain-source and gate-source voltages, respectively, and V_T is the threshold voltage, which can be expressed as⁷

$$V_T = E_{ref} - \psi_0 + \chi^{sol} - \frac{\phi_{si}}{q} - \frac{Q_{it} + Q_f + Q_B}{C_{OX}} + 2\phi_f, \quad (3)$$

where E_{ref} is the reference electrode potential relative to a vacuum, $(-\psi_0 + \chi^{sol})$ describes the interfacial potential at the electrolyte/gate electrode interface (the factor χ^{sol} is the surface dipole moment of the solution, which can be considered constant), $\frac{\phi_{si}}{q}$ is the silicon electron work function, Q_{it} , Q_f , and Q_B are the charge of the interface traps, the fixed oxide charge, and the bulk depletion charge per unit area, respectively, and ϕ_f is the Fermi potential difference between the doped bulk silicon and the intrinsic silicon. Considering the MIP membrane on the gate electrode of the FET, the capacitance and charge in the MIP membrane should be added to equation 3, and can be expressed as

$$V_T = E_{ref} - \psi_0 + \chi^{sol} - \frac{\phi_{si}}{q} - \frac{Q_{it} + Q_f + Q_B + Q_{MIP}}{C_{Com}} + 2\phi_f \quad (4)$$

$$\text{with } C_{Com} = \frac{C_{OX} \cdot C_{MIP}}{C_{OX} + C_{MIP}} = \frac{C_{OX}}{1 + \frac{C_{OX}}{C_{MIP}}}, \quad (5)$$

where Q_{MIP} is the charge in the MIP membrane and C_{Com} is the combined capacitance of C_{OX} and the MIP membrane (C_{MIP}) on the gate electrode. Assuming that C_{MIP} would hardly change after the addition of targeted molecules,^{44,92} C_{Com} is nearly constant regardless of the adsorption of biomolecules, especially small molecules, because C_{OX} is also constant. Moreover, the interfacial potential ($\Delta\psi_0$) at the electrolyte/gate electrode

interface should not change because the ionic concentration (i.e., pH) is basically maintained by using a buffer solution. Also, E_{ref} , $\frac{\phi_{\text{si}}}{q}$, Q_{it} , Q_{f} , Q_{B} , and ϕ_{f} should be the same before and after the molecular recognition events at the MIP interface. Thus, the signal response obtained using a FET sensor is based on the change in V_{T} (ΔV_{T}); therefore, ΔQ_{MIP} should be evaluated in this study on the basis of equation 4 and the above considerations.

The binding affinity of PBA to a diol is pH-dependent, but it is generally understood that the $\text{B}(\text{OH})_3^-$ complex is much stabler than the $\text{B}(\text{OH})_2$ complex.⁹³ For the reversible interaction between the target diol biomolecule (T) and PBA in the MIP membrane (**Figure 3b**),

the rate of formation of the $\text{T} \cdot \text{PBA}^-$ complex at time t is written as

$$\frac{d[\text{T} \cdot \text{PBA}^-]}{dt} = k_{\text{a}}[\text{T}][\text{PBA}] - k_{\text{d}}[\text{T} \cdot \text{PBA}^-], \quad (7)$$

where k_{a} is the association rate constant and k_{d} is the dissociation rate constant. At time t , $[\text{PBA}] = [\text{PBA}]_0 - [\text{T} \cdot \text{PBA}^-]$, where $[\text{PBA}]_0$ is the concentration of PBA at $t = 0$. This is substituted into equation 7 to give

$$\frac{d[\text{T} \cdot \text{PBA}^-]}{dt} = k_{\text{a}}[\text{T}]([\text{PBA}]_0 - [\text{T} \cdot \text{PBA}^-]) - k_{\text{d}}[\text{T} \cdot \text{PBA}^-]. \quad (8)$$

Here, the charge Q_{MIP} is derived from reaction 6; therefore, it is proportional to the formation of the $\text{T} \cdot \text{PBA}^-$ complex in the MIP membrane. Additionally, Q_{max} is proportional to the concentration of PBA in the MIP membrane ($[\text{PBA}]_0$ at $t = 0$), which indicates the capacity of the immobilized ligand. Therefore, equation 8 is modified to

$$\frac{dQ_{\text{MIP}}}{dt} = k_{\text{a}}[c](Q_{\text{max}} - Q_{\text{MIP}}) - k_{\text{d}}Q_{\text{MIP}} = k_{\text{a}}[c]Q_{\text{max}} - (k_{\text{a}}[c] + k_{\text{d}})Q_{\text{MIP}}, \quad (9)$$

where $\frac{dQ_{\text{MIP}}}{dt}$ is the rate of formation of the associated complex ($\text{T} \cdot \text{PBA}^-$) in the MIP membrane (on the gate surface) and $[c]$ is the concentration of the analyte (T) in the solutions. Moreover, integrating equation 9 gives

$$Q_{\text{MIP}}^t = \frac{k_{\text{a}}[c]Q_{\text{max}}[1 - e^{-(k_{\text{a}}[c] + k_{\text{d}})t}]}{k_{\text{a}}[c] + k_{\text{d}}} = \frac{[c]Q_{\text{max}}}{[c] + 1/K_{\text{a}}} [1 - e^{-(K_{\text{a}}[c] + 1)t}], \quad (10)$$

where K_{a} is the binding constant of T and PBA ($k_{\text{a}}/k_{\text{d}}$) in the MIP membrane. From equation 10, $Q_{\text{MIP}}^{t=0} = 0$. Considering equation 4,

$$\Delta V_{\text{T}}(\propto -\Delta V_{\text{out}}) = -\frac{\Delta Q_{\text{MIP}}^t}{C_{\text{Com}}} = -\frac{[c]\Delta V_{\text{out}}^{\text{max}}}{[c] + \frac{1}{K_{\text{a}}}} [1 - e^{-(K_{\text{a}}[c] + 1)t}] \approx -\frac{[c]\Delta V_{\text{out}}^{\text{max}}}{[c] + 1/K_{\text{a}}}, \quad (11)$$

which is estimated after a certain reaction time t . Here, $\Delta V_{\text{out}}^{\text{max}}$ is the maximum change in interfacial potential induced by ΔQ_{max} , which is proportional to the number of binding sites. In this case, ΔV_{out} at the gate was measured at a constant I_{DS} using the source follower circuit.⁸ Therefore, the detected ΔV_{out} is regarded as the change in V_{GS} , which is proportional to $-\Delta V_{\text{T}}$ at a constant I_{DS} .

According to the above considerations, the electrical signal in the entire FET circuit should obey the Langmuir adsorption model. By modifying equation 1 in accordance with equation 11, one can obtain the adsorption isotherm equations for the MIP-FET system as

$$\Delta V_{\text{out}} = \frac{\Delta V_{\text{out}}^{\text{max}} [c]}{1/K_{\text{a}} + [c]}, \quad (12)$$

where $[c]$ is determined as the concentration of the target biomolecule at equilibrium, which is obtained from the saturated electrical signal in real-time measurement. Moreover, the homogeneity and heterogeneity of binding sites distributed in MIPs are critical to effectively enhancing selectivity. Depending on the polymerization processes, the binding sites in MIPs are assumed to be heterogeneously distributed because of the randomness of copolymerization and the template/functional monomer interaction,⁹⁴ therefore, MIPs may include both nonselective and highly selective binding sites at a certain ratio. In this case, the bi-Langmuir adsorption isotherm equation (equation 13) can be utilized for the heterogeneous binding model of the MIP-FET system, instead of equation 12.

$$\Delta V_{\text{out}} = \frac{\Delta V_{1_out}^{\text{max}} [c]}{1/K_{\text{a}1} + [c]} + \frac{\Delta V_{2_out}^{\text{max}} [c]}{1/K_{\text{a}2} + [c]} \quad (13)$$

Equation 13 assumes two main types of binding site with different affinities in the heterogeneous MIP membrane. However, K_{a} can be simply compared among MIPs for various target biomolecules using equation 12, as far as the results analyzed in most previous works are concerned.” (see pages 11 to 15 in the revised and highlighted manuscript)

“In addition, stiffer MIP matrices may be more selective; that is, it should be beneficial to prepare flexible polymeric membranes as rigid as possible.⁹⁵ Actually, electroactive polymeric membranes with relatively high Young’s moduli such as polypyrrole (PPy) and poly(*o*-phenylenediamine) (PoPDA) appear to be utilized as the MIP matrices, preserving the macromolecular arrangements after the template extraction.^{96–102} On the other hand, 2-hydroxyethylmethacrylate (HEMA), ethylene glycol dimethacrylate (EGDMA), and so forth are copolymerized to improve the hydrophilicity of MIP matrices with flexibility, which include small biomolecules and electrolytes and may show

relatively low Young's moduli, when PBA derivatives (e.g., 4-vinyl-PBA) are copolymerized in the MIP matrices on the gate electrode (**Figure 3b**).^{44,50–53} (see page 15 in the revised and highlighted manuscript)

“Note that the MIP matrices should be crosslinked to maintain the cavity size and shape, which are associated with the specificity to and selectivity for target biomolecules.” (see page 15 in the revised and highlighted manuscript)

“In the case of the FET biosensors, particularly, the swelling–deswelling behavior of such hydrogels after the reaction with target biomolecules may cause the change in the capacitance of MIP membranes and then cancel out the electrical signals on the basis of the change in the density of molecular charges. Therefore, the density of cross-linking in the MIP matrices on the gate electrode should be controlled to minimize the swelling–deswelling behavior,^{53,92} although such capacitive signals may be useful for the detection of target biomolecules.” (see page 16 in the revised and highlighted manuscript)

“At pH 7.4, the binding constant for glucose (K_a^{glucose}) with the glucose-selective MIP-FET was estimated as $1.2 \times 10^3 \text{ M}^{-1}$ on the basis of the Langmuir isotherm equation (**Figure 3c**), which was approximately 260 times higher than that of the pristine PBA–glucose binding (4.6 M^{-1}), whereas the binding constant for fructose (K_a^{fructose}) in the glucose-selective MIP ($2.2 \times 10^2 \text{ M}^{-1}$) hardly changed from that for the pristine PBA–fructose binding ($1.6 \times 10^2 \text{ M}^{-1}$).⁷⁹” (see page 16 in the revised and highlighted manuscript)

“ $S_{\text{glucose/fructose}}^{\text{MIP}} = K_a^{\text{glucose}}/K_a^{\text{fructose}} = 5.6$.⁵³ From these calculations, the detection selectivity for glucose to fructose using the glucose-selective MIP-FET was about 200 times higher than that in the pristine PBA–sugar binding ($S_{\text{glucose/fructose}}^{\text{pristine}} = 2.9 \times 10^{-2}$). In this glucose-selective MIP-FET, moreover, LOD was determined to be approximately 3 mM for glucose detection on the basis of the Kaiser limit theory,¹⁰⁹” (see page 16 in the revised and highlighted manuscript)

“Here, LOD would be associated with K_a in equation 12. When ΔV_{out} at the concentration $[c]_{\text{LOD}}$ is $\Delta V_{\text{out}}^{\text{LOD}}$, equation 12 is modified to

$$\Delta V_{\text{out}}^{\text{LOD}} = \frac{\Delta V_{\text{out}}^{\text{max}} [c]_{\text{LOD}}}{1/K_a + [c]_{\text{LOD}}}. \quad (14)$$

Then, equation 14 is rearranged as

$$\log_{10} [c]_{\text{LOD}} = -\log_{10} K_a + \log_{10} \frac{\Delta V_{\text{out}}^{\text{LOD}}}{\Delta V_{\text{out}}^{\text{max}} - \Delta V_{\text{out}}^{\text{LOD}}}. \quad (15)$$

Actually, the glucose-selective MIP-FET shown above sufficiently satisfies equation 15. Furthermore, this trend can be also found in other MIPs, regardless of the readout

technologies (**Figure 3d**).^{52–54,97,98,100,101,106–108,111–115} Therefore, K_a is a parameter used to control LOD. Note that K_a of the target/MIP interaction may be inherently derived from that of the target/functional monomer (e.g., PBA–diol compounds and host–guest interactions). Moreover, as another parameter, the thickness of MIP membranes appears to be related to the LOD (**Figure 3e**).^{48,49,52,53,98,100,108,113,115–118} Thin film MIPs with a thickness less than ca. 50 nm would contribute to the enhancement of LOD, which may result from a high K_a . This means that thicker MIP membranes probably include more heterogeneous sites for binding to target biomolecules.⁵¹ The thickness and adhesiveness of MIP membranes at substrates can be precisely controlled to make them thinner (<50 nm) by some grafting methods such as surface-initiated atom transfer radical polymerization (SI-ATRP), which results in a higher K_a .^{52,113}

On the other hand, a nonimprinted polymer (NIP) should be prepared on the gate electrode as a control polymer by the same method as that for MIP except for adding a target biomolecule as the template. Even NIPs may show some affinities with target biomolecules owing to their nonspecific adsorptions and other properties, resulting in a high K_a . The imprinting factor (IF) is often evaluated as one of the imprinting parameters for target biomolecules, $IF = K_a^{MIP}/K_a^{NIP}$.^{48,52,95,97,102,106,110,112,115,116} That is, a higher IF indicates better MIP performance. For instance, a non-glucose-selective NIP–FET against the glucose-selective MIP–FET mentioned above hardly showed glucose responsivity within the relatively wide range of concentrations, the K_a^{NIP} of which was difficult to analyze;⁵³ that is, K_a^{NIP} was low, resulting in a very high IF. This may be because the NIP suppresses not only the uptake of glucose molecules into itself owing to the higher cross-linking density but also the nonspecific adsorptions owing to its hydrophilicity based on HEMA. Note consider that a PBA-containing hydrogel-coated FET with the lower cross-linking density, the composition of which was almost the same that of the non-glucose-selective NIP-FET, showed some electrical signals for the change in glucose concentrations.⁹² Thus, the K_a values of MIPs and NIPs are the important parameters for controlling S , LOD and IF in the chemically synthesized electrical interface. The number of studies on MIPs is increasing yearly (**Figure S1**); their further applications to FET biosensors are expected in the future, the number of which is still small (**Figure S2**).” (see pages 16 to 18 in the revised and highlighted manuscript)

In 3.1. Polymeric nanofilter-based structural interfaces:

“A Au thin film is used as the gate electrode of an extended-Au-gate–FET (EG-Au–FET), in which the gate electrode is separated and extended from the metal gate of FET because

probe molecules are easily immobilized on the Au gate by $-S-Au$ binding and so forth.”
(see page 19 in the revised and highlighted manuscript)

“Considering the above concept of a structural polymeric nanofilter interface, the relatively thicker MIPs mentioned in section 2.2 may also function as the nanofilter at a distance from the gate surface over the Debye length. Moreover, they supply electrical charges based on the interaction with target biomolecules to the FET biosensor, which are captured in the cavities in the vicinity of the gate surface. This means that the thinner MIP membrane grafted by SI-ATRP on the gate surface should effectively contribute to the generation of electrical signals of the FET biosensor based on target biomolecules without any loss of undetected signals captured in it.” (see page 22 in the revised and highlighted manuscript)

In 3.2. Aptamer nanofilter-based structural interfaces:

“On the other hand, aptamers are utilized as a receptor on electrode surfaces to specifically and selectively trap and detect target biomolecules by overcoming the Debye length limitation with the FET biosensors.⁵⁵ The key point in small-biomarker sensing with the aptamer-based FET biosensor is that the change in the molecular structure of the aptamer is caused by its binding to small biomarkers on the gate surface. For example, the negatively charged backbones of aptamers with a stem-loop structure are expected to move near the gate surface owing to structural reorientation based on the selective binding of the target biomolecule, which means that the negative charges of aptamers are assumed to enter the diffusion layer (i.e., the Debye length) that is less affected by counterions, resulting in the generation of electrical signals. Whether aptamers are utilized as a receptor to trap and detect target biomolecules at the electrode surface or to trap but disable interfering species in the nanofilters, we need to consider the Debye length limit. That is, not only should the ionic strength in the measurement solutions be controlled to be lower or higher, but the thickness of the anchor layer should also be designed to be thinner or thicker.” (see page 23 in the revised and highlighted manuscript)

8. In addition to the comparative listing of advancements in MIPs, it is equally important to provide similar discussions on other vertical sensing modalities.

(Reply)

Thank you very much for this helpful suggestion. In accordance with the reviewer’s comment, we have added **Figures 3d** and **3e** in the revised manuscript to precisely discuss the correlation between LOD and K_a or thickness on MIPs combined with various readout

technologies, instead of **Table 1** (see pages 16 and 17 in the revised and highlighted manuscript).

9. In addition to emphasizing their significance, the figures in the review should be visually appealing, avoiding excessive white space and the use of overly vibrant colors.

(Reply)

Thank you very much for this helpful suggestion. In accordance with the reviewer's comment, we have modified all the figures and added new ones.

10. The overall content of the article is relatively limited, with some discussions being excessively broad and shallow. In addition to literature references, the author should provide more original insights and prospects for the development of the field. More constructive guidance and recommendations on limitations and challenges would be highly valuable.

(Reply)

Thank you very much for this helpful suggestion. We have thoroughly modified the manuscript and particularly provided the original insights and prospects for the development of the field in accordance with the other reviewers' comments (from Reviewers 1 to 3). We would be grateful if the reviewer could understand our revisions through our Replies 1 to 9 shown above and our replies to the other reviewers' comments.

[Reviewer #3]

Comments: In this manuscript, authors aim to highlight recent advances in the electrochemical interfaces which are one of the key issues for biosensors. In particular, the diverse signal transduction interfaces are summarized for enzyme-free electrochemical biosensor for example, chemically synthesized, physically or chemically structured interfaces. These features are important for controlling biosensing performance such as selectivity, sensitivity, limit of detection or S/N ratio). Considering these issues, this paper is interesting in this area, and is well organized and beneficial for the researchers or readers of this journal. However, there are some weaknesses to be addressed before publication. I recommend the major revision for this manuscript. My questions are as follows:

1. It is suggested to add more paragraph with respect to the field-effect transistor in the introduction.

(Reply)

Thank you very much for this helpful suggestion. In accordance with the reviewer's comment, we have added some introductory information on the FET biosensor, including its principle, advantages, and so forth, as follows. In accordance with this revision, we have modified the references. In particular, **Figure 1b** has been added to the revised manuscript.

In 1. Introduction:

“A platform based on a solution-gated field-effect transistor (FET), which originates from electronics, is suitable for use in miniaturized and cost-effective systems to directly measure biological samples as the FET biosensor in the field of *in vitro* diagnostics.¹ Such miniaturized electronic devices can be easily equipped with a wireless function and attached to the body, which are available for wearable biosensors to detect biomarkers in tears, sweat, and saliva, that is, for diagnostics in a blood-sampling free manner.²⁻⁴ In general, the gate insulator surface (e.g., SiO₂) is directly in contact with a measurement solution in the FET biosensor without a metal gate electrode, which is different from a metal-oxide-semiconductor (MOS) transistor, for which the potential of the measurement solution is controlled by the reference electrode,⁵⁻⁷ as shown in **Figure 1a**. When ions or biomolecules with charges are adsorbed on the gate insulator surface, their charges electrostatically interact with electrons across the gate insulator, resulting in a change in the conductivity of the channel of the FET. That is, the drain-source current (I_{DS}) at the channel changes with the change in the density of ions or biomolecules with charges adsorbed on the gate insulator. That is, such charged species induce a change in the interfacial potential (V_{out}) of the solution/gate insulator at a constant I_{DS} , which is potentiometrically detected.^{5,8} Oxide and nitride membranes used as the gate insulator and the passivation layer can effectively detect a change in pH on the basis of the reaction in equilibrium between hydrogen ions and hydroxy groups at their surfaces.⁵ Afterwards, various ion-sensitive membranes (ISMs)^{2,9-15} and biomimetic receptors with enzymes, antibodies, and single-stranded DNAs¹⁶⁻²¹ were coated on the gate electrode surfaces as the FET biosensors to specifically and selectively detect target ions and biomolecules, and further cellular activities were monitored on the basis of the change in ionic behaviors at the cell/gate interface in real time.²²⁻²⁵ This is because the detection principle of FET biosensors is basically derived from the potentiometric measurement of the changes in ionic and biomolecular charges or membrane capacitances at the electrolyte solution/gate electrode interface. Moreover, various semiconductive materials have been widely utilized as the channel of FETs for biosensing devices, such as one-dimensional [1D (e.g., nanotubes and nanowires)] and two-dimensional [2D (e.g., graphene, diamond, and

MoS₂)] materials.^{15,26–30} In particular, a solution-gated 1D or 2D-channel FET biosensor, the channel of which is directly in contact with an electrolyte solution, is expected to have a steep subthreshold slope (SS), resulting in an ultrahighly sensitive biosensing, owing to a relatively large capacitance of the electric double-layer at the electrolyte solution/channel interface.^{27,31} Moreover, thin-film transistors (TFTs) such as transparent amorphous oxide semiconductors (e.g., amorphous In–Ga–Zn–oxide) can be applied as one of the FET biosensors, which are deposited on transparent substrates such as glass and plastics.³² Thus, the number of studies on the FET biosensors is increasing yearly (**Figure 1b**).” (see pages 3 to 4 in the revised and highlighted manuscript)

2. It is suggested to modify the Abstract to highlight the innovation of this paper.

(Reply)

Thank you very much for this helpful suggestion. In accordance with the reviewer’s comment, we have modified Abstract to clarify this point in this manuscript, as follows.

Abstract:

“A platform of a field-effect transistor (FET) biosensor based on electronics is suitable for use in miniaturized and cost-effective systems that are required in the field of *in vitro* diagnostics. This enables the direct detection of ionic or biomolecular charges in a biosample, which contributes to label-free biosensing. In addition, various semiconductive materials have been applied as the channel of FETs for biosensing, such as one- and two-dimensional materials. Thus, studies on FET biosensors are intensively pursued, the number of which is rapidly increasing. Here, a signal transduction interface material between the biosample and the channel of FETs plays a key role in capturing target ions or biomolecules, which then induces their electrochemical reactions into output signals. A versatile concept for a signal transduction interface is required for various biomarkers because there are no enzymes or antibodies applicable to every target biomarker. In this review, distinctive signal transduction interfaces for the FET biosensors are introduced, such as chemically synthesized, physically structured, and biologically induced interfaces, without relying on enzymes or antibodies. Diverse signal transduction interfaces in the FET biosensors become a key element in controlling biosensing parameters, such as specificity, selectivity, binding constant, limit of detection, signal-to-noise ratio, and biocompatibility.” (see page 2 in the revised and highlighted manuscript)

3. The advantage/disadvantages in each section (from 2.1 to 4) should be briefly discussed in the text.

(Reply)

Thank you very much for this helpful suggestion. We have added some discussions in each section, as follows. In particular, **Figures 3d, 3e, S1, and S2** have been added in the revised manuscript.

In 1. Introduction:

“The Debye length λ depends on the ionic strength of the electrolyte solution used and is expressed as $\lambda = (\epsilon_0 \epsilon_r k_B T / 2 N_A e^2 I)^{1/2}$, where I is the ionic strength of the electrolyte solution, ϵ_0 is the permittivity of free space, ϵ_r is the dielectric constant, k_B is the Boltzmann constant, T is the absolute temperature, N_A is the Avogadro number, and e is the elementary charge. The Debye length limit is controlled by changing the ionic strength of a measurement solution, that is, diluted measurement solutions are useful for improving the detection sensitivity of the FET biosensors to charged biomolecules because of the reduction of the shielding effect by counterions.” (see page 6 in the revised and highlighted manuscript)

In 2.1. Ion-sensitive membranes and their biocompatibility:

“Moreover, the selectivity of a crown ether L for ions M_1^{n+} and M_2^{n+} is expressed as the ratio of each binding constant $K_a(1) / K_a(2)$; e.g., $K_a(1) = [M_1 L^{n+}] / ([M_1^{n+}] + [L])$. The complexing ability, that is, K_a of crown ether with ions, is dependent on the relative sizes of the cavity of the crown ether and the ions.⁸⁰ In addition, poly- and bis (crown ether)s are favorable for the formation of 2:1 (crown ether unit/ion) sandwich-type complexes with ions, which contribute to the increase in K_a , that is, the selectivity to a specific ion.⁸¹” (see page 9 in the revised and highlighted manuscript)

“Hydrophilic polydopamine (poly-DA) modification actually enhanced the anti-adhesive properties of ISMs by increasing the surface hydrophilicity.⁸⁶ This is why a poly-DA film prevented the nonspecific adsorptions of proteins that may generate noise signals on the ISMs, while the detection sensitivity for monovalent ions did not deteriorate, keeping it near the Nernstian response. Thus, the modified ISM-based FETs can be utilized more safely and precisely in actual biological samples with interfering species as wearable biosensors with flexibility.” (see page 10 in the revised and highlighted manuscript)

In 2.2. MIP-based electrical interfaces:

“For instance, glucose molecules used as the template, which bind to PBA, are removed under acidic conditions, whereas glucose boronate esters induce negative charges in the MIP matrices under relatively basic conditions, because the pKa of a glucose boronate ester is 6.8.⁷⁹” (see page 11 in the revised and highlighted manuscript)

“To understand the chemical basis of interactions between the MIP and the target biomolecules underlying the electrical responses of MIP-based FET biosensors, quantitative analysis is required. In general, the characteristics of the binding of a target molecule to MIP are quantified using adsorption isotherm equations, as the binding process involves the reversible adhesion of the target molecule to the target-selective membrane.⁹¹ In this way, the potentiometric analyses based on the FET biosensor can directly characterize the MIP interface without the batch rebinding process, which is often required for MIP characterization. According to a previous study,⁹¹ the Langmuir adsorption isotherm equation for a bulk rebinding system is expressed as

$$B = \frac{N[c]}{1 + K_a[c]}, \quad (1)$$

where B refers to a signal observed at equilibrium for the MIP-bound template, $[c]$ to the free concentration of the template at equilibrium, N to the number of available active centers in the MIP per unit volume, and K_a to the binding constant. Equation 1 assumes homogeneously distributed binding sites with a constant binding constant K_a .

The operation of a silicon-based FET in the unsaturated region can generally be described as

$$I_{DS} = \mu C_{OX} \frac{W}{L} \left[(V_{GS} - V_T) V_{DS} - \frac{1}{2} V_{DS}^2 \right], \quad (2)$$

where μ is the electron mobility in the channel, C_{OX} is the gate oxide capacitance, $\frac{W}{L}$ is the channel width-to-length ratio, V_{DS} and V_{GS} are the applied drain–source and gate–source voltages, respectively, and V_T is the threshold voltage, which can be expressed as⁷

$$V_T = E_{ref} - \psi_0 + \chi^{sol} - \frac{\phi_{si}}{q} - \frac{Q_{it} + Q_f + Q_B}{C_{OX}} + 2\phi_f, \quad (3)$$

where E_{ref} is the reference electrode potential relative to a vacuum, $(-\psi_0 + \chi^{sol})$ describes the interfacial potential at the electrolyte/gate electrode interface (the factor χ^{sol} is the surface dipole moment of the solution, which can be considered constant), $\frac{\phi_{si}}{q}$ is the silicon electron work function, Q_{it} , Q_f , and Q_B are the charge of the interface traps, the fixed oxide charge, and the bulk depletion charge per unit area, respectively, and ϕ_f is the Fermi potential difference between the doped bulk silicon and the intrinsic silicon.

Considering the MIP membrane on the gate electrode of the FET, the capacitance and charge in the MIP membrane should be added to equation 3, and can be expressed as

$$V_T = E_{\text{ref}} - \psi_0 + \chi^{\text{sol}} - \frac{\phi_{\text{si}}}{q} - \frac{Q_{\text{it}} + Q_{\text{f}} + Q_{\text{B}} + Q_{\text{MIP}}}{C_{\text{Com}}} + 2\phi_{\text{f}} \quad (4)$$

$$\text{with } C_{\text{Com}} = \frac{C_{\text{OX}} \cdot C_{\text{MIP}}}{C_{\text{OX}} + C_{\text{MIP}}} = \frac{C_{\text{OX}}}{1 + \frac{C_{\text{OX}}}{C_{\text{MIP}}}}, \quad (5)$$

where Q_{MIP} is the charge in the MIP membrane and C_{Com} is the combined capacitance of C_{OX} and the MIP membrane (C_{MIP}) on the gate electrode. Assuming that C_{MIP} would hardly change after the addition of targeted molecules,^{44,92} C_{Com} is nearly constant regardless of the adsorption of biomolecules, especially small molecules, because C_{OX} is also constant. Moreover, the interfacial potential ($\Delta\psi_0$) at the electrolyte/gate electrode interface should not change because the ionic concentration (i.e., pH) is basically maintained by using a buffer solution. Also, E_{ref} , $\frac{\phi_{\text{si}}}{q}$, Q_{it} , Q_{f} , Q_{B} , and ϕ_{f} should be the same before and after the molecular recognition events at the MIP interface. Thus, the signal response obtained using a FET sensor is based on the change in V_T (ΔV_T); therefore, ΔQ_{MIP} should be evaluated in this study on the basis of equation 4 and the above considerations.

The binding affinity of PBA to a diol is pH-dependent, but it is generally understood that the $\text{B}(\text{OH})_3^-$ complex is much stabler than the $\text{B}(\text{OH})_2$ complex.⁹³ For the reversible interaction between the target diol biomolecule (T) and PBA in the MIP membrane (**Figure 3b**),

the rate of formation of the $\text{T} \cdot \text{PBA}^-$ complex at time t is written as

$$\frac{d[\text{T} \cdot \text{PBA}^-]}{dt} = k_a[\text{T}][\text{PBA}] - k_d[\text{T} \cdot \text{PBA}^-], \quad (7)$$

where k_a is the association rate constant and k_d is the dissociation rate constant. At time t , $[\text{PBA}] = [\text{PBA}]_0 - [\text{T} \cdot \text{PBA}^-]$, where $[\text{PBA}]_0$ is the concentration of PBA at $t = 0$. This is substituted into equation 7 to give

$$\frac{d[\text{T} \cdot \text{PBA}^-]}{dt} = k_a[\text{T}]([\text{PBA}]_0 - [\text{T} \cdot \text{PBA}^-]) - k_d[\text{T} \cdot \text{PBA}^-]. \quad (8)$$

Here, the charge Q_{MIP} is derived from reaction 6; therefore, it is proportional to the formation of the $\text{T} \cdot \text{PBA}^-$ complex in the MIP membrane. Additionally, Q_{max} is proportional to the concentration of PBA in the MIP membrane ($[\text{PBA}]_0$ at $t = 0$), which indicates the capacity of the immobilized ligand. Therefore, equation 8 is modified to

$$\frac{dQ_{\text{MIP}}}{dt} = k_a[c](Q_{\text{max}} - Q_{\text{MIP}}) - k_d Q_{\text{MIP}} = k_a[c]Q_{\text{max}} - (k_a[c] + k_d)Q_{\text{MIP}}, \quad (9)$$

where $\frac{dQ_{\text{MIP}}}{dt}$ is the rate of formation of the associated complex ($\text{T} \cdot \text{PBA}^-$) in the MIP membrane (on the gate surface) and $[c]$ is the concentration of the analyte (T) in the solutions. Moreover, integrating equation 9 gives

$$Q_{\text{MIP}}^t = \frac{k_a[c]Q_{\text{max}}[1 - e^{-(k_a[c] + k_d)t}]}{k_a[c] + k_d} = \frac{[c]Q_{\text{max}}}{[c] + 1/K_a} [1 - e^{-(K_a[c] + 1)t}], \quad (10)$$

where K_a is the binding constant of T and PBA (k_a/k_d) in the MIP membrane. From equation 10, $Q_{\text{MIP}}^{t=0} = 0$. Considering equation 4,

$$\Delta V_{\text{T}} (\propto -\Delta V_{\text{out}}) = -\frac{\Delta Q_{\text{MIP}}^t}{C_{\text{Com}}} = -\frac{[c]\Delta V_{\text{out}}^{\text{max}}}{[c] + \frac{1}{K_a}} [1 - e^{-(K_a[c] + 1)t}] \approx -\frac{[c]\Delta V_{\text{out}}^{\text{max}}}{[c] + 1/K_a}, \quad (11)$$

which is estimated after a certain reaction time t . Here, $\Delta V_{\text{out}}^{\text{max}}$ is the maximum change in interfacial potential induced by ΔQ_{max} , which is proportional to the number of binding sites. In this case, ΔV_{out} at the gate is measured at a constant I_{DS} using the source follower circuit.⁸ Therefore, the detected ΔV_{out} is regarded as the change in V_{GS} , which was proportional to $-\Delta V_{\text{T}}$ at a constant I_{DS} .

According to the above considerations, the electrical signal in the entire FET circuit should obey the Langmuir adsorption model. By modifying equation 1 in accordance with equation 11, one can obtain the adsorption isotherm equations for the MIP-FET system as

$$\Delta V_{\text{out}} = \frac{\Delta V_{\text{out}}^{\text{max}} [c]}{1/K_a + [c]}, \quad (12)$$

where $[c]$ is determined as the concentration of the target biomolecule at equilibrium, which is obtained from the saturated electrical signal in real-time measurement. Moreover, the homogeneity and heterogeneity of binding sites distributed in MIPs are critical to effectively enhancing selectivity. Depending on the polymerization processes, the binding sites in MIPs are assumed to be heterogeneously distributed because of the randomness of copolymerization and the template/functional monomer interaction;⁹⁴ therefore, MIPs may include both nonselective and highly selective binding sites at a certain ratio. In this case, the bi-Langmuir adsorption isotherm equation (equation 13) can be utilized for the heterogeneous binding model of the MIP-FET system, instead of equation 12.

$$\Delta V_{\text{out}} = \frac{\Delta V_{1_out}^{\text{max}} [c]}{1/K_{a1} + [c]} + \frac{\Delta V_{2_out}^{\text{max}} [c]}{1/K_{a2} + [c]} \quad (13)$$

Equation 13 assumes two main types of binding site with different affinities in the heterogeneous MIP membrane. However, K_a can be simply compared among MIPs for various target biomolecules using equation 12, as far as the results analyzed in most previous works are concerned.” (see pages 11 to 15 in the revised and highlighted manuscript)

“In addition, stiffer MIP matrices may be more selective; that is, it should be beneficial to prepare flexible polymeric membranes as rigid as possible.⁹⁵ Actually, electroactive polymeric membranes with relatively high Young’s moduli such as polypyrrole (PPy) and poly(*o*-phenylenediamine) (PoPDA) appear to be utilized as the MIP matrices, preserving the macromolecular arrangements after the template extraction.^{96–102} On the other hand, 2-hydroxyethylmethacrylate (HEMA), ethylene glycol dimethacrylate (EGDMA), and so forth are copolymerized to improve the hydrophilicity of MIP matrices with flexibility, which include small biomolecules and electrolytes and may show relatively low Young’s moduli, when PBA derivatives (e.g., 4-vinyl-PBA) are copolymerized in the MIP matrices on the gate electrode (**Figure 3b**).^{44,50–53}” (see page 15 in the revised and highlighted manuscript)

“Note that the MIP matrices should be crosslinked to maintain the cavity size and shape, which are associated with the specificity to and selectivity for target biomolecules.” (see page 15 in the revised and highlighted manuscript)

“In the case of the FET biosensors, particularly, the swelling–deswelling behavior of such hydrogels after the reaction with target biomolecules may cause the change in the capacitance of MIP membranes and then cancel out the electrical signals on the basis of the change in the density of molecular charges. Therefore, the density of cross-linking in the MIP matrices on the gate electrode should be controlled to minimize the swelling–deswelling behavior,^{53,92} although such capacitive signals may be useful for the detection of target biomolecules.” (see pages 15 and 16 in the revised and highlighted manuscript)

“At pH 7.4, the binding constant for glucose (K_a^{glucose}) with the glucose-selective MIP-FET was estimated as $1.2 \times 10^3 \text{ M}^{-1}$ on the basis of the Langmuir isotherm equation (**Figure 3c**), which was approximately 260 times higher than that of the pristine PBA–glucose binding (4.6 M^{-1}), whereas the binding constant for fructose (K_a^{fructose}) in the glucose-selective MIP ($2.2 \times 10^2 \text{ M}^{-1}$) hardly changed from that for the pristine PBA–fructose binding ($1.6 \times 10^2 \text{ M}^{-1}$).⁷⁹” (see page 16 in the revised and highlighted manuscript)

“ $S_{\text{glucose/fructose}}^{\text{MIP}} = K_a^{\text{glucose}} / K_a^{\text{fructose}} = 5.6$.⁵³ From these calculations, the detection selectivity for glucose to fructose using the glucose-selective MIP-FET was about 200 times higher than that in the pristine PBA–sugar binding ($S_{\text{glucose/fructose}}^{\text{pristine}} = 2.9 \times 10^{-2}$). In this glucose-

selective MIP-FET, moreover, LOD was determined to be approximately 3 mM for glucose detection on the basis of the Kaiser limit theory,¹⁰⁹ (see page 16 in the revised and highlighted manuscript)

“Here, LOD would be associated with K_a in equation 12. When ΔV_{out} at the concentration $[c]_{LOD}$ is ΔV_{out}^{LOD} , equation 12 is modified to

$$\Delta V_{out}^{LOD} = \frac{\Delta V_{out}^{max} [c]_{LOD}}{1/K_a + [c]_{LOD}}. \quad (14)$$

Then, equation 14 is rearranged as

$$\log_{10}[c]_{LOD} = -\log_{10}K_a + \log_{10} \frac{\Delta V_{out}^{LOD}}{\Delta V_{out}^{max} - \Delta V_{out}^{LOD}}. \quad (15)$$

Actually, the glucose-selective MIP-FET shown above sufficiently satisfies equation 15. Furthermore, this trend can be also found in other MIPs, regardless of the readout technologies (**Figure 3d**).^{52–54,97,98,100,101,106–108,111–115} Therefore, K_a is a parameter used to control LOD. Note that K_a of the target/MIP interaction may be inherently derived from that of the target/functional monomer (e.g., PBA–diol compounds and host–guest interactions). Moreover, as another parameter, the thickness of MIP membranes appears to be related to the LOD (**Figure 3e**).^{48,49,52,53,98,100,108,113,115–118} Thin film MIPs with a thickness less than ca. 50 nm would contribute to the enhancement of LOD, which may result from a high K_a . This means that thicker MIP membranes probably include more heterogeneous sites for binding to target biomolecules.⁵¹ The thickness and adhesiveness of MIP membranes at substrates can be precisely controlled to make them thinner (<50 nm) by some grafting methods such as surface-initiated atom transfer radical polymerization (SI-ATRP), which results in a higher K_a .^{52,113}

On the other hand, a nonimprinted polymer (NIP) should be prepared on the gate electrode as a control polymer by the same method as that for MIP except for adding a target biomolecule as the template. Even NIPs may show some affinities with target biomolecules owing to their nonspecific adsorptions and other properties, resulting in a high K_a . The imprinting factor (IF) is often evaluated as one of the imprinting parameters for target biomolecules, $IF = K_a^{MIP}/K_a^{NIP}$.^{48,52,95,97,102,106,110,112,115,116} That is, a higher IF indicates better MIP performance. For instance, a non-glucose-selective NIP–FET against the glucose-selective MIP–FET mentioned above hardly showed glucose responsivity within the relatively wide range of concentrations, the K_a^{NIP} of which was difficult to analyze;⁵³ that is, K_a^{NIP} was low, resulting in a very high IF. This may be because the NIP suppresses not only the uptake of glucose molecules into itself owing to the higher cross-linking density but also the nonspecific adsorptions owing to its hydrophilicity based on HEMA. Note consider that a PBA-containing hydrogel-coated FET with the

lower cross-linking density, the composition of which was almost the same that of the non-glucose-selective NIP-FET, showed some electrical signals for the change in glucose concentrations.⁹² Thus, the K_a values of MIPs and NIPs are the important parameters for controlling S , LOD and IF in the chemically synthesized electrical interface. The number of studies on MIPs is increasing yearly (**Figure S1**); their further applications to FET biosensors are expected in the future, the number of which is still small (**Figure S2**).” (see pages 16 to 18 in the revised and highlighted manuscript)

In 3.1. Polymeric nanofilter-based structural interfaces:

“A Au thin film is used as the gate electrode of an extended-Au-gate-FET (EG-Au-FET), in which the gate electrode is separated and extended from the metal gate of FET because probe molecules are easily immobilized on the Au gate by $-S-Au$ binding and so forth.” (see page 19 in the revised and highlighted manuscript)

“Considering the above concept of a structural polymeric nanofilter interface, the relatively thicker MIPs mentioned in section 2.2 may also function as the nanofilter at a distance from the gate surface over the Debye length. Moreover, they supply electrical charges based on the interaction with target biomolecules to the FET biosensor, which are captured in the cavities in the vicinity of the gate surface. This means that the thinner MIP membrane grafted by SI-ATRP on the gate surface should effectively contribute to the generation of electrical signals of the FET biosensor based on target biomolecules without any loss of undetected signals captured in it.” (see page 22 in the revised and highlighted manuscript)

In 3.2. Aptamer nanofilter-based structural interfaces:

“On the other hand, aptamers are utilized as a receptor on electrode surfaces to specifically and selectively trap and detect target biomolecules by overcoming the Debye length limitation with the FET biosensors.⁵⁵ The key point in small-biomarker sensing with the aptamer-based FET biosensor is that the change in the molecular structure of the aptamer is caused by its binding to small biomarkers on the gate surface. For example, the negatively charged backbones of aptamers with a stem-loop structure are expected to move near the gate surface owing to structural reorientation based on the selective binding of the target biomolecule, which means that the negative charges of aptamers are assumed to enter the diffusion layer (i.e., the Debye length) that is less affected by counterions, resulting in the generation of electrical signals. Whether aptamers are utilized as a receptor to trap and detect target biomolecules at the electrode surface or to trap but disable interfering species in the nanofilters, we need to consider the Debye length limit.

That is, not only should the ionic strength in the measurement solutions be controlled to be lower or higher, but the thickness of the anchor layer should also be designed to be thinner or thicker.” (see page 23 in the revised and highlighted manuscript)

4. The more advanced functional interfaces, its detailed description of the sensing mechanism, advantage and disadvantages should be clearly presented. In particular, the key challenges are not presented.

(Reply)

In this manuscript, we intend to introduce the more advanced functional interfaces through our studies. In particular, various semiconductor materials have recently been applied as the channel of FET biosensors. Therefore, the diversity of signal transduction interfaces broadens the possibility of developing novel biosensing devices, in parallel with the development of new channel materials for the FET biosensors. For example, the number of studies on MIPs is increasing yearly (**Figure S1**); their further applications to FET biosensors are expected in the future, the number of which is still small (**Figure S2**). Since we have thoroughly revised the manuscript, we hope that the concept of this review article is now clearer to the reviewer.

5. It is recommended that the authors should add more references along with the detailed sensing performances such as sensitivity, S/N ratio, limit of detection in Table 1, not just focusing in only MIP-based sensor. In addition, if applicable, the advantages and disadvantages of each biosensor would be good in the Table 1.

(Reply)

Thank you very much for this helpful suggestion. In accordance with the reviewer’s comment, we have reconsidered the references; 24 references shown in the original manuscript have been deleted, and 36 references have been added in the revised manuscript. In particular, by citing more references, we have added **Figures 3d** and **3e** in the revised manuscript to precisely discuss the correlation between LOD and K_a or thickness on MIPs combined with various readout technologies, instead of presenting the advantages or disadvantages in **Table 1**. In this review article, we intend to introduce the advanced functional interfaces for the FET biosensors in an enzyme-free manner and have commented on how the change in the density of charges based on biomolecular recognition events is directly transduced into electrical signals at the signal transduction interfaces on the basis of the detection principle of FET biosensors. That is, presenting the diverse signal transduction interfaces, such as the chemically synthesized, physically and chemically structured, and biologically induced interfaces, will broaden the

possibility of developing novel biosensing devices, in parallel with the development of new channel materials of the FET biosensors, rather than clearly showing the advantages/disadvantages of each concept. We would be grateful if the reviewer could understand our concept in this review article. Therefore, we have added the following sentences in Conclusion and outlook.

“In particular, the increase in K_a for the target biological target, which results in the enhancement of S and LOD, becomes a key challenge for enzyme-free interfaces, and then biocompatible materials may be chosen for the signal transduction interfaces. On the other hand, various semiconductor materials have been recently applied as the channel of FET biosensors. Therefore, the diversity of signal transduction interfaces broadens the possibility of developing novel biosensing devices, in parallel with the development of new channel materials of the FET biosensors.” (see pages 27 to 28 in the revised and highlighted manuscript)

“Therefore, the diversity of signal transduction interfaces broadens the possibility of developing novel biosensing devices, in parallel with the development of new channel materials of the FET biosensors. In addition, the steep SS, which is one of the transistor characteristics, should lead to the increase in the sensitivity of biosensing, although it is a key parameter of FETs (*see* 1. Introduction),^{27,31} and then the optimal biosensing parameters are effectively provided by the diverse signal transduction interfaces for the practical use of FET biosensors.” (see page 28 in the revised and highlighted manuscript)

6. It would be better to the readers when author could change the Figure 1 with whole schematic diagram showing different interface, characteristics and their corresponding detection method for the target detection.

(Reply)

Thank you very much for this helpful suggestion. In accordance with the reviewer’s comment, we have modified not only **Figure 1** but also all the figures in the revised manuscript, corresponding to all revisions.

7. The outlook viewpoints or key challenges should be presented in the conclusion section. I would recommend to re-write this conclusion section.

(Reply)

Thank you very much for this helpful suggestion. In accordance with the reviewer’s comment, we have modified Conclusion and outlook, as follows.

Conclusion and outlook:

“In developing a biosensor, we consider the design criteria based on its three components, namely, the biological target, signal transduction interface, and detection device. Among the detection devices, a platform based on an electronic device with the FET biosensors is suitable for use in miniaturized and cost-effective systems to directly measure biological samples because the FET biosensors enable the direct detection of intrinsic ionic and biomolecular charges in principle, which contributes to label- and enzyme-free biosensing. Such miniaturized electronic devices can be easily equipped with a wireless function and attached to the body, which are available for wearable biosensors to detect biomarkers in a blood-sampling free manner (i.e., tears, sweat, and saliva). Here, it is very important to determine how the change in the density of charges based on biomolecular recognition events is directly transduced into electrical signals at the signal transduction interface, regardless of the wearability of FET biosensors. Such bio/device interfaces are chemically synthesized, physically and chemically structured, and biologically induced to control the biosensing parameters such as specificity, S , K_a , LOD, S/N, and biocompatibility with respect to the biological target, although the chemically synthesized electrical interfaces are also useful as the signal transduction interfaces for the wearable biosensors with flexibility. In particular, the increase in K_a for the target biological target, which results in the enhancement of S and LOD, becomes a key challenge for enzyme-free interfaces, and then biocompatible materials may be chosen for the signal transduction interfaces. On the other hand, various semiconductor materials have been recently applied as the channel of FET biosensors. Therefore, the diversity of signal transduction interfaces broadens the possibility of developing novel biosensing devices, in parallel with the development of new channel materials of the FET biosensors. In addition, the steep SS, which is one of the transistor characteristics, should lead to the increase in the sensitivity of biosensing, although it is a key parameter of FETs (*see* 1. Introduction),^{27,31} and then the optimal biosensing parameters are effectively provided by the diverse signal transduction interfaces for the practical use of FET biosensors.” (see pages 27 to 28 in the revised and highlighted manuscript)

8. Figure2 should be changed because this is little confusing.

(Reply)

Thank you very much for this helpful suggestion. In accordance with the reviewer’s comment, we have modified not only **Figure 2** but also all the figures in the revised manuscript.

9. The section 3.2 in the text should be re-written.

(Reply)

Thank you very much for this helpful suggestion. In accordance with the reviewer's comment, we have modified Section 3.2 and particularly added the following sentences in the revised manuscript.

“On the other hand, aptamers are utilized as a receptor on electrode surfaces to specifically and selectively trap and detect target biomolecules by overcoming the Debye length limitation with the FET biosensors.⁵⁵ The key point in small-biomarker sensing with the aptamer-based FET biosensor is that the change in the molecular structure of the aptamer is caused by its binding to small biomarkers on the gate surface. For example, the negatively charged backbones of aptamers with a stem–loop structure are expected to move near the gate surface owing to structural reorientation based on the selective binding of the target biomolecule, which means that the negative charges of aptamers are assumed to enter the diffusion layer (i.e., the Debye length) that is less affected by counterions, resulting in the generation of electrical signals. Whether aptamers are utilized as a receptor to trap and detect target biomolecules at the electrode surface or to trap but disable interfering species in the nanofilters, we need to consider the Debye length limit. That is, not only should the ionic strength in the measurement solutions be controlled to be lower or higher, but the thickness of the anchor layer should also be designed to be thinner or thicker.” (see pages 23 in the revised and highlighted manuscript)

10. It would be better that the comparison of 3 different interfaces described in the text (advantage/disadvantage, insight or key issue) is presented as another Table.

(Reply)

In this review article, we intend to introduce the advanced functional interfaces for the FET biosensors in an enzyme-free manner and have commented on how the change in the density of charges based on biomolecular recognition events is directly transduced into electrical signals at the signal transduction interfaces on the basis of the detection principle of FET biosensors. That is, the diverse signal transduction interfaces such as the chemically synthesized, physically and chemically structured, and biologically induced interfaces broaden the possibility of developing novel biosensing devices, in parallel with the development of new channel materials of the FET biosensors, rather than clearly showing the advantages/disadvantages of each concept. We would be grateful if the reviewer could understand our concept in this review article. (see Reply 5)

11. There are some typographical errors in the reference section. It means that the first word should start with a capital letter. Please address accordingly.

(Reply)

Thank you very much for this helpful suggestion. In accordance with the reviewer's comment, we have thoroughly checked and modified the manuscript.

REVIEWERS' COMMENTS:

Reviewer #1 (Remarks to the Author):

[Editorial Note: This reviewer has not provided any further comments for the authors.]

Reviewer #2 (Remarks to the Author):

On the basis of the initial version, the author has made substantial improvements and progress in the manuscript, particularly in terms of structural clarity and the depth of the literature review. The presentation of concepts and discussion is more refined. These enhancements not only elevate the overall quality and readability of the paper but also address the reviewer's suggestions effectively. The author's diligent response to the reviewer's comments is commendable.

On the other hand, it is still recommended that the author continue refining the manuscript to ensure alignment with the journal's guidelines for a comprehensive and impactful review article. Here are a few pre-publication suggestions:

1. The text in Figure 1a becomes difficult to discern after rotation and should be presented in an alternative manner. Additionally, the use of italics and underlining should be more cautious to avoid impacting overall readability.
2. The annotation text in Figure 1b overlaps with the border lines, suggesting repositioning to enhance font clarity.
3. Some of the text in Figures 3a and 3b is too small. If it is essential, consider enlarging it; if not, consider removing it for clarity. The same situation applies to Figure 4 as well.
4. Considering the widespread use of semiconductor materials in the channels of FET biosensors, the diversity of signal transduction interfaces mentioned by the author opens up broad possibilities for the development of novel biosensing devices. Therefore, discussing the challenges and limitations of different interfaces in specific scenarios would contribute to a more comprehensive understanding of the field.

Reviewer #3 (Remarks to the Author):

This reviewer think that the author have carefully revised the manuscript accordin to the some comments of the reviewers and been greatly improved the quality. so it can be accepted to publish of Communications Chemistry.

[Reviewer #1]

[Editorial Note: This reviewer has not provided any further comments for the authors.]

(Reply 1)

Thank you very much for your kind review.

[Reviewer #2]

Comments:

On the basis of the initial version, the author has made substantial improvements and progress in the manuscript, particularly in terms of structural clarity and the depth of the literature review. The presentation of concepts and discussion is more refined. These enhancements not only elevate the overall quality and readability of the paper but also address the reviewer's suggestions effectively. The author's diligent response to the reviewer's comments is commendable. On the other hand, it is still recommended that the author continue refining the manuscript to ensure alignment with the journal's guidelines for a comprehensive and impactful review article. Here are a few pre-publication suggestions:

1. The text in Figure 1a becomes difficult to discern after rotation and should be presented in an alternative manner. Additionally, the use of italics and underlining should be more cautious to avoid impacting overall readability.

(Reply)

Thank you very much for pointing this out. In accordance with the reviewer's comment, we have revised Figure 1a.

2. The annotation text in Figure 1b overlaps with the border lines, suggesting repositioning to enhance font clarity.

(Reply)

Thank you very much for pointing this out. In accordance with the reviewer's comment, we have revised Figure 1b.

3. Some of the text in Figures 3a and 3b is too small. If it is essential, consider enlarging it; if not, consider removing it for clarity. The same situation applies to Figure 4 as well.

(Reply)

Thank you very much for pointing this out. In accordance with the reviewer's comment, we have revised Figures 3 and 4.

4. Considering the widespread use of semiconductor materials in the channels of FET biosensors, the diversity of signal transduction interfaces mentioned by the author opens up broad possibilities for the development of novel biosensing devices. Therefore, discussing the challenges and limitations of different interfaces in specific scenarios would contribute to a more comprehensive understanding of the field.

(Reply)

Thank you very much for pointing this out. In accordance with the reviewer's comment, we have added the following comments in the revised manuscript (Conclusion and outlook).

“In addition, the FET biosensors can be applied to semiconductor integrated circuits to measure multiple samples simultaneously. This is one of the advantages of utilizing semiconductor technology and also the unique feature of FETs because other biosensors (e.g., surface plasmon resonance and quartz crystal microbalance sensors) hardly enable the integration of electrodes as in complementary metal oxide semiconductor sensors.²¹ That is, it will be a challenge to coat and arrange different signal transduction interface materials for various biomarkers on the individual gate electrodes in the arrayed devices in the future. In this case, an enormous quantity of detected data is assumed including complicated information; therefore, the data analysis based on a bioinformatics method may be desirable according to the omics approach.” (page 29 in the revised manuscript)

[Reviewer #3]

Comments:

This reviewer think that the author have carefully revised the manuscript according to the some comments of the reviewers and been greatly improved the quality. so it can be accepted to publish of Communications Chemistry.

(Reply)

Thank you very much for your kind review.